# *In silico* study of alkaloids with quercetin nucleus for inhibition of SARS-CoV-2 protease and receptor cell protease

**Ali Mohebbi**[1]*, **Marzieh Eskandarzadeh**[2], **Hanieh Zangi**[1], **Marzie Fatehi**[1]

**1** Department of Chemical Engineering, Faculty of Engineering, Shahid Bahonar University of Kerman, Kerman, Iran, **2** Research Committee of Faculty of Pharmacy, Lorestan University of Medical Science, Khorramabad, Iran

* amohebbi@uk.ac.ir, amohebbi2002@yahoo.com

**Data Availability Statement:** All relevant data are within the manuscript.

**Funding:** The author(s) received no specific funding for this work.

## Abstract

Covid-19 disease caused by the deadly SARS-CoV-2 virus is a serious and threatening global health issue declared by the WHO as an epidemic. Researchers are studying the design and discovery of drugs to inhibit the SARS-CoV-2 virus due to its high mortality rate. The main Covid-19 virus protease (Mpro) and human transmembrane protease, serine 2 (TMPRSS2) are attractive targets for the study of antiviral drugs against SARS-2 coronavirus. Increasing consumption of herbal medicines in the community and a serious approach to these drugs have increased the demand for effective herbal substances. Alkaloids are one of the most important active ingredients in medicinal plants that have wide applications in the pharmaceutical industry. In this study, seven alkaloid ligands with Quercetin nucleus for the inhibition of Mpro and TMPRSS2 were studied using computational drug design including molecular docking and molecular dynamics simulation (MD). Auto Dock software was used to evaluate molecular binding energy. Three ligands with the most negative docking score were selected to be entered into the MD simulation procedure. To evaluate the protein conformational changes induced by tested ligands and calculate the binding energy between the ligands and target proteins, GROMACS software based on AMBER03 force field was used. The MD results showed that Phyllospadine and Dracocephin-A form stable complexes with Mpro and TMPRSS2. Prolinalin-A indicated an acceptable inhibitory effect on Mpro, whereas it resulted in some structural instability of TMPRSS2. The total binding energies between three ligands, Prolinalin-A, Phyllospadine and Dracocephin-A and two proteins MPro and TMRPSS2 are (-111.235 ± 15.877, - 75.422 ± 11.140), (-107.033 ± 9.072, -84.939 ± 10.155) and (-102.941 ± 9.477, - 92.451 ± 10.539), respectively. Since the binding energies are at a minimum, this indicates confirmation of the proper binding of the ligands to the proteins. Regardless of some Prolinalin-A-induced TMPRSS2 conformational changes, it may properly bind to TMPRSS2 binding site due to its acceptable binding energy. Therefore, these three ligands can be promising candidates for the development of drugs to treat infections caused by the SARS-CoV-2 virus.

**Competing interests:** The authors have declared that no competing interests exist.

## Introduction

Coronaviruses are a group of viruses belonging to the Coronaviridae family that play an essential role in developing respiratory diseases. These viruses can cause the common cold and pneumonia or bronchitis [1–3]. Coronaviruses contain a positive single-stranded RNA genome, ranging from approximately 29 to 32 kilobases. The newest type of coronavirus is called coronavirus 2 (SARS-CoV-2), the causative agent of COVID-19, which is a highly contagious viral disease [4, 5]. Its mRNAs encode four structural proteins named spike glycoprotein (S), envelope (E), membrane (M), and nucleocapsid (N). The N protein is located in the viral genome structure, M and E are responsible for viral gathering, and S is the most important one that is located on the viral membrane and composed of two subunits $S_1$ and $S_2$, involved in the attachment and membrane fusion of the virus, respectively [6–9]. This glucoprotein needs a proteolytic priming to be activated to mediate SARS-CoV-2 entry in the host cell. The tentacles of glycoprotein S bind the virus to angiotensin-converting enzyme 2 (ACE2 (receptor of the host cell [10, 11], which has been recognized as the primary receptor for SARS-CoV-2 spike protein. The ACE2 receptor is mostly found in pulmonary capillaries, it is also found in other sites, but its main accumulation site is the pulmonary capillaries endothelium [12]. Additionally, transmembrane protease, serine 2 (TMPRSS2) is a human type II intermembrane serine protease that has been shown its role in conducting the priming of SARS-CoV-2 spike glycoprotein by procreating two distinct segments of the viral spike protein, $S_1/S_2$ and $S_2$' [6, 13]. Therefore, these host's respiratory tract proteins have a pivotal role in the mechanisms of cell entry and viral infection, then they can be an attractive drug target in SARS-CoV-2 therapeutics [14–17]. Besides, the main protease (Mpro) of SARS-CoV-2 is a cysteine protease with a molecular weight of 33.8 kD which plays a key role in the evolution of the functional proteins, the transcription and replication of SARS-CoV-2 genome [18, 19]. Hence, the neutralization of Mpro is an inhibition method to prevent the replication of viral RNA in the host cell. Therefore, Mpro is a potential drug target against coronaviruses [20–22]. The structure of SARS-CoV-2 Mpro is made of three domains (I-III), which are highly conserved among coronaviruses. In this protease, the substrate binding site is located between domains I and II, and there is an unusual catalytic Cys-His dyad site in the gap between these two domains [23–25]. The cysteine residue of the Cys-His pair undergoes a nucleophilic attack on the substrate-binding active site. Around this pair, Mpro forms a stable binding pocket consisting of four subsites $S_1$, $S_1$', $S_2$, and $S_4$, which are well adapted to the substrate [25–27].

Alkaloids with quercetin nuclei and potential inhibitory activity can be considered an effective medicine against SARS-CoV-2 [28–30]. Additionally, these compounds have important biological properties, including antimicrobial, anticancer, anti-inflammation, and analgesic activities [31–35]. Many derivatives of alkaloids have different effects on enzymes based on their substitutions. Seven derivatives of alkaloids are suggested as inhibitors in this research, namely A-Davallioside, D-Dracocephin, ProlinalinA, Phyllospadine, Lilaline, Drahebephin-A, and Dracocephin-A.

Since this disease has led to the death of many humans and no effective medicine has been discovered for its treatment, studies are focused on the design and discovery of medicines that can inhibit SARS- CoV -2. Mpro of the Covid-19 and the host cell protease (TMPRSS2) are attractive targets for the study of antiviral agents against SARS-CoV-2. Drug discovery using computational methods gives new insights into understanding protein-ligand interactions virtually. Molecular docking and molecular dynamics (MD) simulations are valuable tools for computational drug discovery. Therefore, the purpose of this work was to perform computational studies including molecular docking and MD simulation to evaluate the mechanism of

Mpro and TMPRSS2 inhibition by seven alkaloid ligands with quercetin nuclei and propose appropriate ligands for developing a drug to treat SARS-CoV-2.

## Computational approaches

### Preparation of the primary structures

The 3D structure of Mpro (PDB ID: 6LU7 in complex with an inhibitor N3-Chain A) and TMPRSS2 (PDB ID: 7MEQ in complex with Nafamostat) proteins were obtained from the protein data bank (PDB) [36]. Afterwards, the primary structure of all selected alkaloids used in the study was drawn by Avogadro software [37] (Fig 1). Then, the energy of constructed structures was minimized using the Consistent Valence Force Field (CVFF) and saved as PDB file (pdb) by using this software.

### Molecular docking

Docking simulation was performed using Autodock 1.5.2.4 software [38]. This step aims to identify the potential ligand binding sites. All possible protein-ligand conformations were implemented in docking by the Lamarckian genetic algorithm (LGA). The protein was

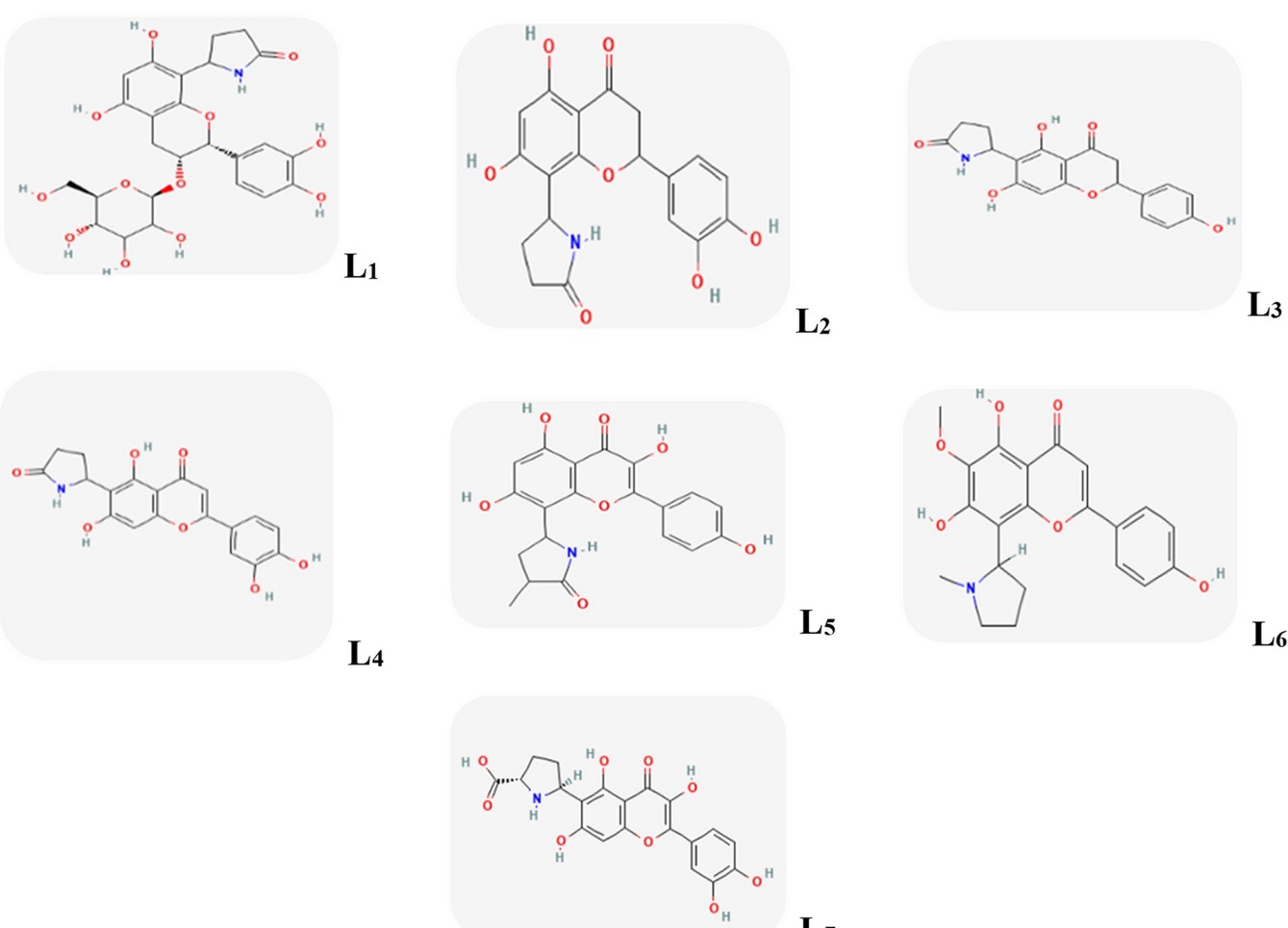

**Fig 1.** L₁: Davallioside-A, L₂: Dracocephin-D, L₃: Dracocephin-A, L₄: Drahebephin-A, L₅: Lilaline, L₆: Phyllospadine, L₇: ProlinalinA.

prepared by removing water molecules, ions, and cognate ligands. Then, polar hydrogen atoms, and Kollman Charge was assigned to the protein. To prepare the ligand, hydrogen atoms were added to the optimized structure at neutral pH, followed by determining the residues involved in the protein binding site. The docking process was calculated by adjusting the coordination of the grid box center in dimensions of 70 Å × 70 Å × 72 Å along the x, y, and z axes. Then flexible ligand docking was done with the number of runs of 100 and docking results were announced in the form of binding energy (kcal/mol).

## Molecular dynamics simulation

At this stage, the protein of interest and the best-tested ligands with the protein were simulated to further examine the effects of ligand binding on the conformation of the protein through molecular dynamics simulation. GROMACS 2019.1 [39] with Ubuntu operating system (version 18.04) Linux on an Intel core 12 Quad 6800k 3.6GhHZ, GPu = stu 1080 ti NVIDIA, and 16GBas was used for this purpose. The topology was constructed using the AMBER03 force field. Then, each system was placed in the center of a cubic box (61 Å × 61 Å× 61 Å) with a minimum distance of 1 nm from the edges of the box under periodic boundary conditions. Water molecules were added to each simulation box, and the SPC model was the selected water model as it is the standard water model of GROMACS software. Since the structure of proteins is usually positively or negatively charged due to the presence of charged amino acids, the charge of the box must be neutralized before simulation. For this purpose, two molecules of $Cl^-$ and $Na^+$ were added to the solution. There are several reasons for system instability, such as the addition of ions and solvents around the complex or the presence of destabilizing interactions. Therefore, the energy minimization step is required as the initial step of the simulation, which prevents the occurrence of errors and stoppage of calculations [40–42]. In this study, energy was minimized in 50,000 steps using the steepest descent algorithm. Then, the simulation was carried out in two stages under the NVT and NpT ensembles with a time step of 2 fs. The Berendsen thermostat and Berendsen barostat were applied to achieve constant temperature and pressure, 300 K and 1 bar, respectively. Also, the cut-off radius was considered to be 1.2 nm. The selected algorithm for NpT and NVT was LINC that also used for production steps to retain the bonds' lengths. Besides, long-range electrostatic interactions were determined by Particle Mesh Ewald (PME) [43]. Coulomb radius and Fourier grid spacing were considered to be 1.2 and 0.16 nm, respectively, and the van der Waals interactions were set at 1.2 nm. Finally, the simulation was done for 100 ns, and all resulted trajectories in MD stage were analyzed using built-in GROMACS services, including radius of gyration (Rg), root mean squared fluctuation (RMSF), root mean squared deviation (RMSD), solvent accessible surface area (SASA and Principal Component Analysis (PCA).

## Calculation of the binding energy

The calculation of the binding energy is important to analyze the binding affinity of inhibitors to the receptors. The g_mmpbsa program was used, which is described as:

$$\Delta G_{binding} = \Delta G_{complex} - (\Delta G_{protein} + \Delta G_{ligand}) \qquad (1)$$

where $\Delta G_{complex}$, $\Delta G_{protein}$ and $\Delta G_{ligand}$ are the total MM-PBSA energy of the protein-ligand complex, solution-free energy of protein, and solution-free energy of ligand, respectively [44, 45].

## Results and discussion

### Molecular docking studies of Mpro and TMPRSS2 proteins in complex with 7 alkaloid ligands

The docking procedure can be the best early step of the computational drug design and discovery method, which reveals the binding affinity and modes between ligand and protein. The current study focuses on the main protease (Mpro/3Clpro) (PDB ID: 6LU7) as one of the target proteins for the inhibition of SARS-CoV-2. The three distinct domains, the active site residues, and other structural features of the Mpro crystal structure are shown in Fig 2. Each protomer contains domain I (residues 8–101), domain II (residues 102–184) and domain III (residues 201–303). The binding site is a cleft between domain I and domain II and Cys-His dyad is considered as the catalytic site residues [22].

To find the lowest possible binding energy, the ligands were attached to SARS-CoV-2 protease. Among the 100 dockings performed for each SARS-CoV-2 protein complex with alkaloid

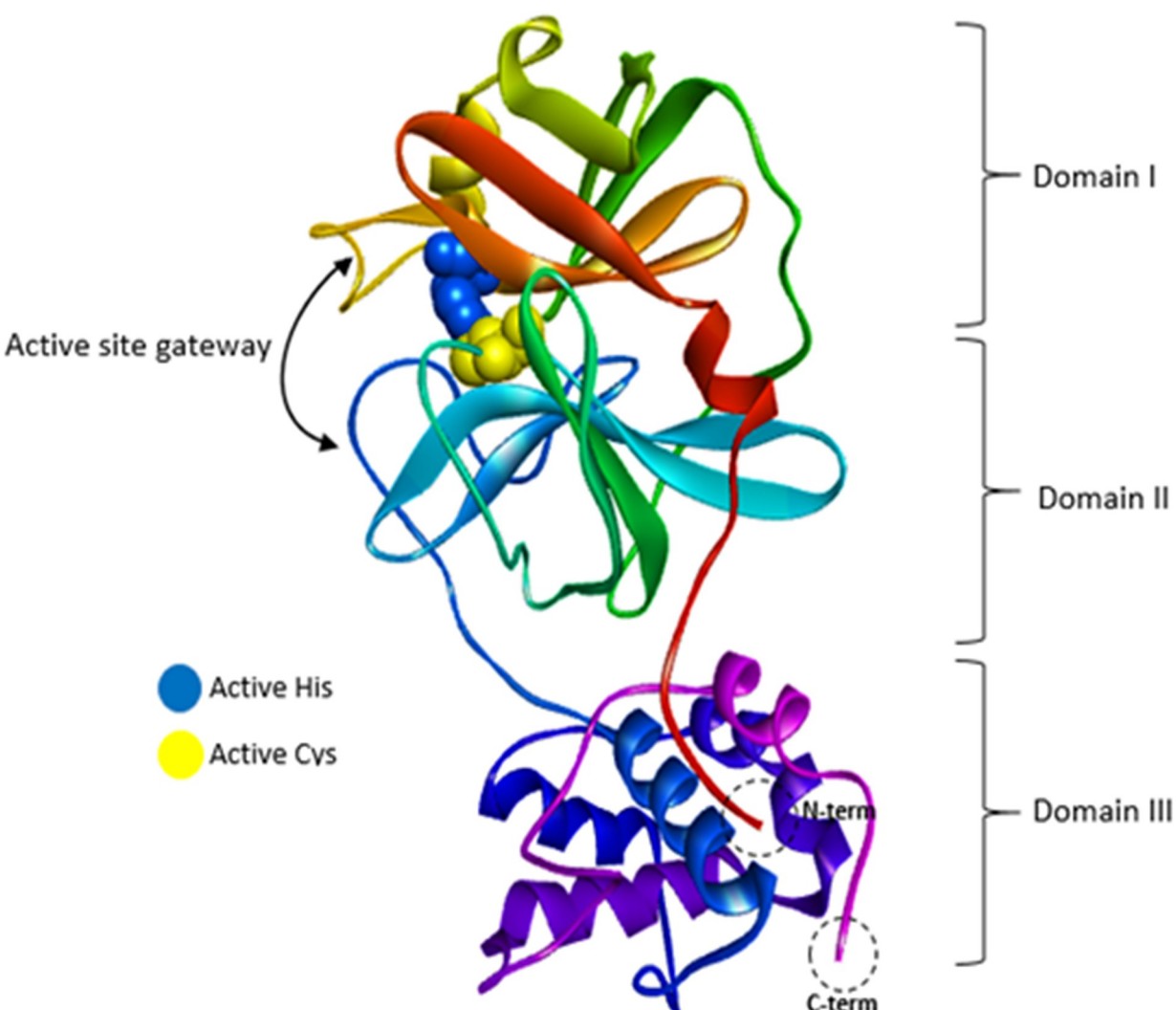

**Fig 2. Ribbon demonstration for the monomer of Mpro with key structural features labeled, including three distinct domains, His41 (Blue) and Cys145 (yellow) residues of the catalytic dyad, and C-terminal and N-terminal.**

**Table 1. Binding free energies of tested ligands with TMPRSS and Mpro.**

| Entry | Docking Score (kcal/mol) of SARS-CoV-2 Mpro (6LU7) | Docking Score (kcal/mol) of Human TMPRSS2 (7MEQ) |
|---|---|---|
| L₁ | -8 | -7.7 |
| L₂ | -8.1 | -7.4 |
| L₃ | -8.3 | -8.1 |
| L₄ | -7.7 | -7.4 |
| L₅ | -7.6 | -6.7 |
| L₆ | -8.6 | -8.2 |
| L₇ | -8.4 | -7.8 |

structures, the conformation with the lowest binding energy was evaluated, which is given for all complexes in Table 1. L₆ had the highest affinity to the Mpro enzyme due to the most negative binding energy of -8.6 kcal/mol in comparison with other ligands. As shown in Fig 3F, His41, and Cys145 as key residues of the catalytic dyad in Mpro [18, 46] had binding interactions with L₆. According to Firouzi et al.'s study [47], Asn142- Ser144 as oxyanion hole residues and residues His164-Glu166 are other critical parts of the substrate-binding site of Mpro. As shown in the 3-D figures (Fig 3F), L₆ strongly bound to the active site of Mpro via forming hydrogen bonds with Glu166, Cys145, Ser144, and π-π interaction with His41 residues that are a non-negligible force for stabilizing porous supramolecular frameworks, while it formed hydrophobic bonds with Met165 and Gln189. All hydrogen, hydrophobic, and van der Waals interactions between the protein and tested ligands, are provided in Table 2. As can be seen, L₇ and L₃ with -8.4 and -8.3 kcal/mol docking scores, respectively, placed in second and third position in terms of binding free energy that their interaction with His41 can be noticed. That is to say, all alkaloid ligands had higher docking scores compared to reference ligand (N3: -6.3 kcal/mol) in complex with Mpro and the hydrogen interactions of N3 are with Gln184, Glu166, His164, and Gly143 residues of Mpro in Fig 4. It seems that flavon structure plays a prominent role in their binding to this protein. As in Saakre et al.'s study, Quercetin as a flavonoid [46] was demonstrated as a potential inhibitor of Mpro that establishes H bonds with LYS88, TYR101, LYS137, GLY138, ASP289, GLY143, and CYS145 [46] and proves the importance of this structure in our alkaloids ligand to interact with catalytic site of Mpro. Moreover, Nguyen et al. [16] discovered the presence of a double bond in position and the presence of hydroxyl group in C-5′ position of B ring in flavonoid skeletons C2–C3, carbonyl group at the C-4 position in the C ring facilitate their hydrogen bonds and the electrostatic interactions with the active site of Mpro. This fact aligns with our results regarding the better docking score of L6 compared to other ligands. Hydrogen bond distance and angle are some of the other criteria to ensure the stability of H bond between protein and ligand. The acceptable ranges of H bond distance between donor and acceptor and a favorable donor–H acceptor angle ranges are 2.4–3.5 Å and 120˚-180˚, respectively [48]. All our docked ligands formed stable hydrogen bonds with Mpro and the most H bonds between L₃, L₆ and L₇ and Mpro were Covalent and strong because their distances were from 2.2 to 2.5 Å.

TMPRSS2 is a transmembrane serine protease that facilitates the entry of coronaviruses into host cells. Therefore, the inhibition of TMPRSS2 can modulate SARS-CoV-2 infection. The extracellular region of TMPRSS2 is composed of three domains including the LDLR class A (residues 112–149), the scavenger receptor cysteine-rich domain 2 (SRCR-2) (residues 150–242) and the Peptidase S1 (residues 256–489), which contains His296, Asp345 and S441 as the protease active site residues [49] (Fig 5). Based on Idris et al.'s study [50], three residues Asp435, Ser460, and Gly462 are the critical residues of substrate-binding pocket and three

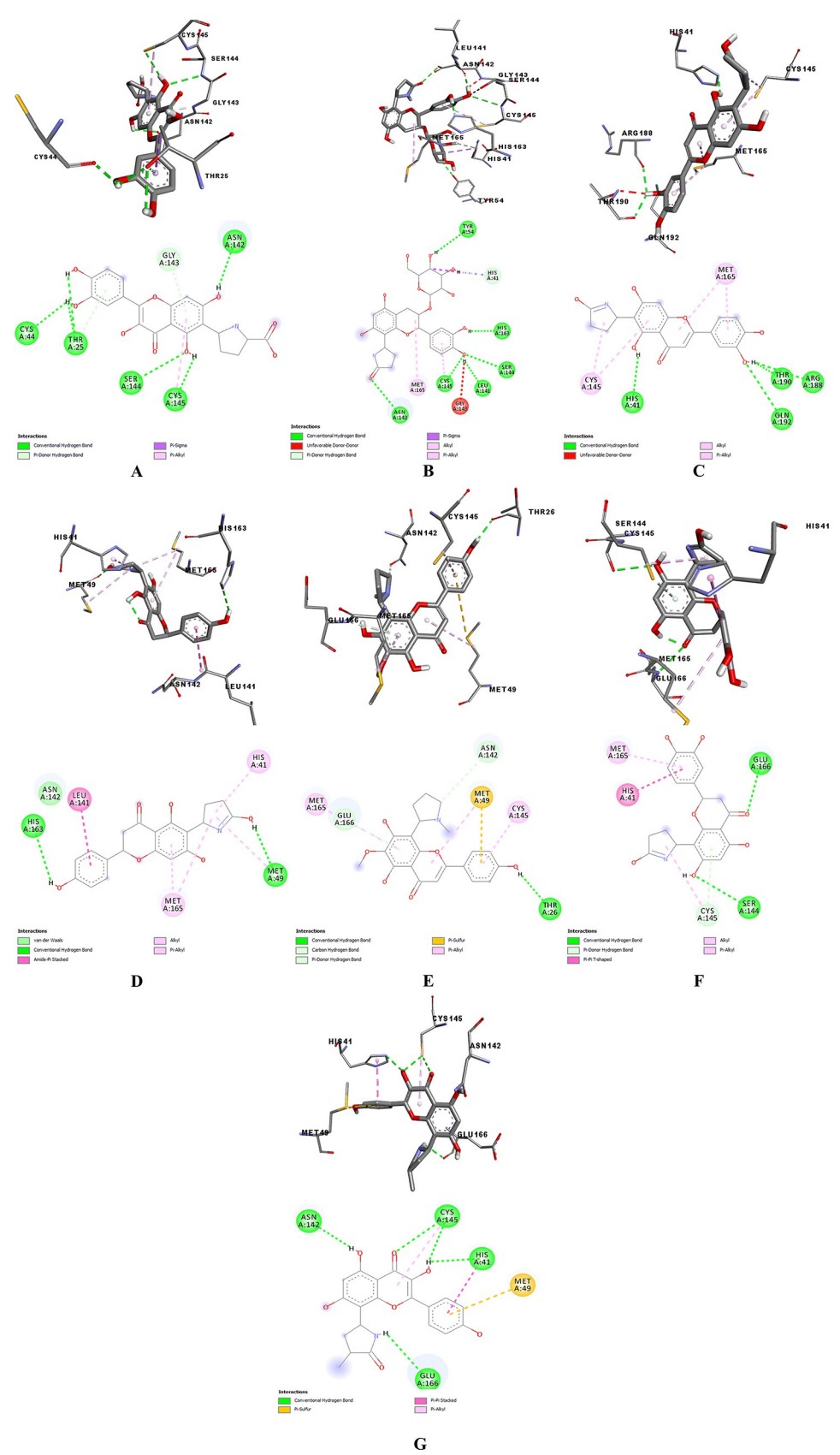

**Fig 3.** 3D (up) and 2D (down) binding interactions of $L_1$ (A), $L_2$ (B), $L_3$ (C), $L_4$ (D), $L_5$ (E), $L_6$ (F), and $L_7$ (G) with Mpro.

residues His296, Asp345, and Ser441 have a key role in the catalytic site of TMPRSS2. Table 1 provides the binding free energy of TMPRSS2 with the seven under-studied ligands. Regarding TMPRSS2-$L_6$ complex, $L_6$ had the strongest binding energy (-8.2 kcal/mol) compared to Nafa-mostat (−5.424 kcal/mol) [51] as a reference standard ligand and other tested ligands in this study. Asp435, Ser436, Gly464, and Gly439 were the residues of TMPRSS2, which formed hydrogen bonds with Nafamostat (Fig 6). $L_6$ as seen in Fig 7F, formed three conventional hydrogen bonds with His296 and two others with Cys465 and Ser441, a carbon hydrogen bond with Ser460, and hydrophobic bonds with His296 residues. Indeed, not only all interac-tions of $L_6$ were established in the active site, but also its optimum hydrogen bonds distance and angle confirm its strength in comparison with other docked ligands. $L_3$ is another ligand with high affinity (-8.1 kcal/mol binding free energy) to TMPRSS2 that interacted with resi-dues His296, and Gln438 by hydrogen bonds and residues Glu299 and Gln438 by hydrophobic bonds. Also, it seems that the higher binding free energy of $L_6$ than $L_3$ is due to the more hydrogen interactions formed by the pyrrolidine structure with TMPRSS2 catalytic site. The pyrrolidine structure can be considered a critical part that plays a key role in the formation of

**Table 2. Molecular binding results of docked compounds with Mpro.**

| MPRO (6LU7) | Hydrogen bond | Distance (Å) | Donor Angle (degree) | Other Interactions |
|---|---|---|---|---|
| $L_1$ | CYS145, | 2.34 | 146.57 | GLY143, CYS145, THR25 |
| | ASN142 | 2.56 | 159.28 | |
| | SER144 | 2.80 | 110.01 | |
| | THR25 | 2.63 | 130.11 | |
| | CYS44 | 2.79 | 123.74 | |
| $L_2$ | HIS41 | 3.19 | 115.08 | GLY143, MET165, CYS145 |
| | TYR54 | 2.53 | 105.14 | |
| | LEU141 | 2.07 | 121.64 | |
| | ASN142 | 2.30 | 130.08 | |
| | SER144 | 2.23 | 132.30 | |
| | CYS145 | 2.72 | 125.44 | |
| | HIS163 | 2.02 | 172.31 | |
| $L_3$ | HIS41 | 2.33 | 126.41 | MET165, CYS145 |
| | THR190 | 2.60 | 128.24 | |
| | ARG188 | 2.46 | 117.42 | |
| | GLN192 | 3.01 | 129.33 | |
| $L_4$ | MET49 | 2.70 | 105.96 | ASN142, LEW141, HIS41, MET165 |
| | HIS163 | 3.04 | 113.23 | |
| $L_5$ | THR26 | 2.24 | 114.64 | MET49, CYS145, ASN142 |
| | GLU166 | 2.43 | 146.00 | |
| $L_6$ | SER144 | 2.08 | 125.93 | MET165, HIS41, CYS145 |
| | CYS145 | 2.44 | 127.51 | |
| | GLU166 | 2.13 | 129.65 | |
| $L_7$ | CYS145 | 2.63 | 141.25 | MET49, HIS41, CYS145 |
| | ASN142 | 2.24 | 160.69 | |
| | HIS41 | 2.51 | 135.46 | |
| | GLU166 | 2.48 | 115.35 | |

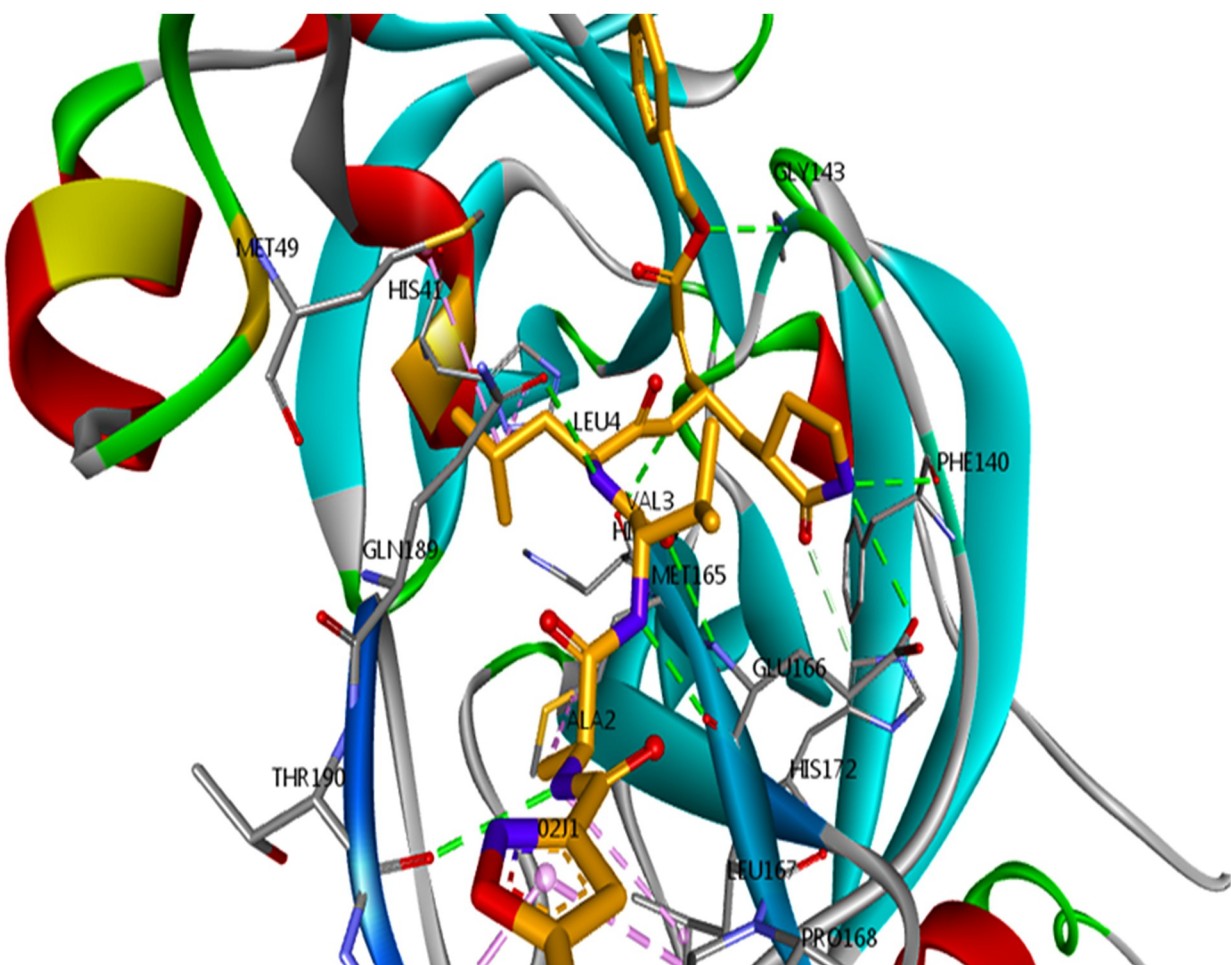

**Fig 4. 3D binding interactions of N3 as reference inhibitor of Mpro, the hydrogen bonds are shown in green dotted lines and residues with hydrophobic interactions placed around the ligand with purple dotted lines.**

stable protein-ligand complexes because of the presence of nitrogen atoms in this structure. Gyebi et al. [52] showed that the amidinonitrogen of guanidine group in Camostat as one of the reference inhibitors of Mpro established hydrogen bonds with the catalytic residues. It affirms the importance of nitrogen in the pyrrolidine structure. The other tested ligands' interactions with TMPRSS2 are listed in Table 3.

## MD simulation of Mpro and TMPRSS2 proteins in complex with selected ligands

While molecular docking is a reliable approach for evaluating ligand-protein interactions in detail, MD simulation was carried out for three of the ligands ($L_3$, $L_6$, and $L_7$) with the most negative binding energy for both proteins to investigate protein conformation changes following the ligand binding to it.

First, to study the range of structural changes and stability of the studied systems and reach the equilibrium state, RMSD was calculated for the main protease, Mpro complexes with $L_3$, $L_6$, and $L_7$ ligands, and TMPRSS2, TMPRSS2 complex with $L_3$, $L_6$, and $L_7$ ligands. Fig 8A depicts $L_7$ and $L_3$ RMSD changed between 0.2 and 0.4 and reached equilibrium after 70 ns.

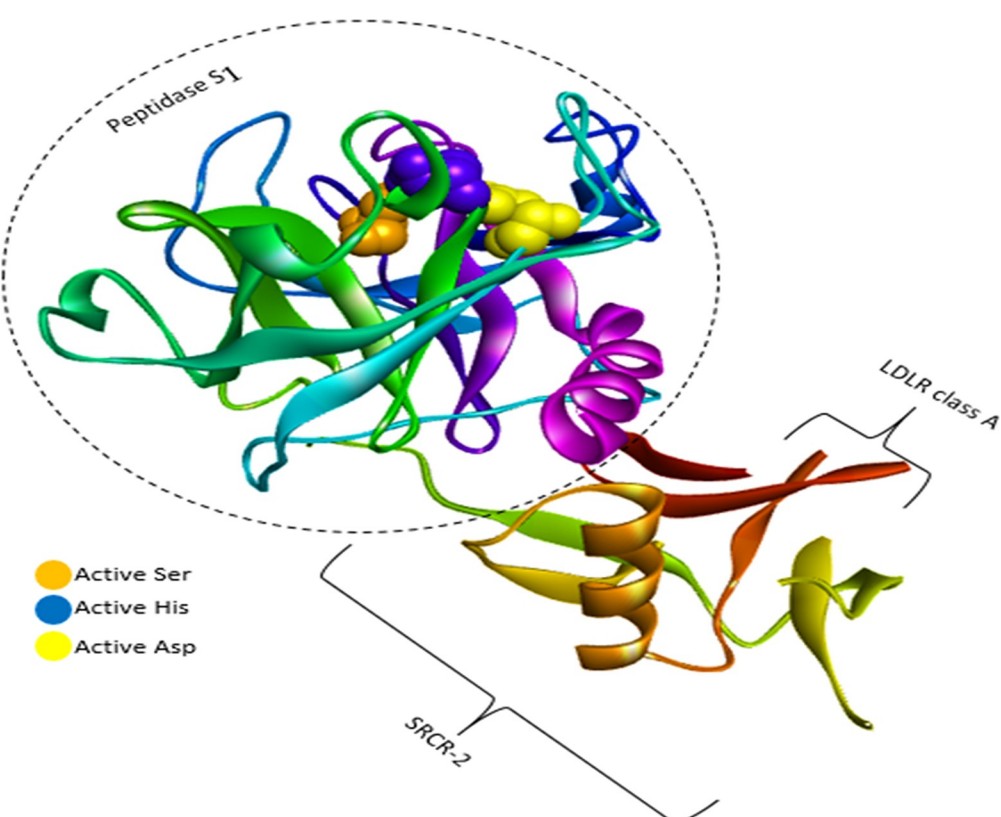

**Fig 5. Ribbon demonstration for the TMPRSS2 with key structural features labeled, including extracellular region (LDLR class A, SRCR-2, Peptidase S$_1$), Ser441 (Orange), His296 (Blue), and Asp 345 residues of the catalytic triad.**

Observed slight fluctuations at various intervals of L$_7$ and L$_3$ were induced by ligand conformation change at the Mpro active site. The last 30 ns RMSD of L$_6$-Mpro complex was particularly similar to Mpro indicating more stability of L$_6$-Mpro complex in comparison with other ligands. The RMSD of L$_6$-Mpro complex is $<$ 0.35 nm and this value is about 0.2nm for Mpro. These results illustrated all selected ligands result in Mpro conformation changes; however, changes induced by L6 binding are negligible because of its low RMSD and the similarity of its changes pattern to Mpro.

For the TMPRSS2 protein and its binding ligands, it was observed that all systems were in equilibrium after 20 ns (Fig 8B). The mean of this value for TMPRSS2, L$_6$, L$_7$, and L$_3$ is partially 0.34, 0.34, 0.8 and 0.42, in turn. L$_6$-TMPRSS2 complex plot is well aligned with TMPRSS2 plot and their similar RMSD reveals system stability during simulation. On the other hand, this figure shows a sudden change in RMSD values of L$_7$-TMPRSS2 complex. This change is not a reason for the system disequilibrium at the beginning of the simulation, although its higher RMSD than L$_6$ and L$_3$ leads to system instability that will be examined by RMSF plot to validate RMSD results.

The RMSF analysis was obtained to investigate the protein's flexible area and the contribution of each amino acid residue to the structural fluctuations of the ligand-protein complex. RMSF data for L$_3$-Mpro, L$_6$-Mpro, and L$_7$-Mpro complexes are shown in Fig 9A. Most of the residues in the Mpro complex form had a fluctuation below 0.2 nm, which indicates that the residues were stable during the molecular dynamic simulation. In addition, the N- and C-terminal residues have undergone large fluctuations. It seems that residues LEU141, TYR 154,

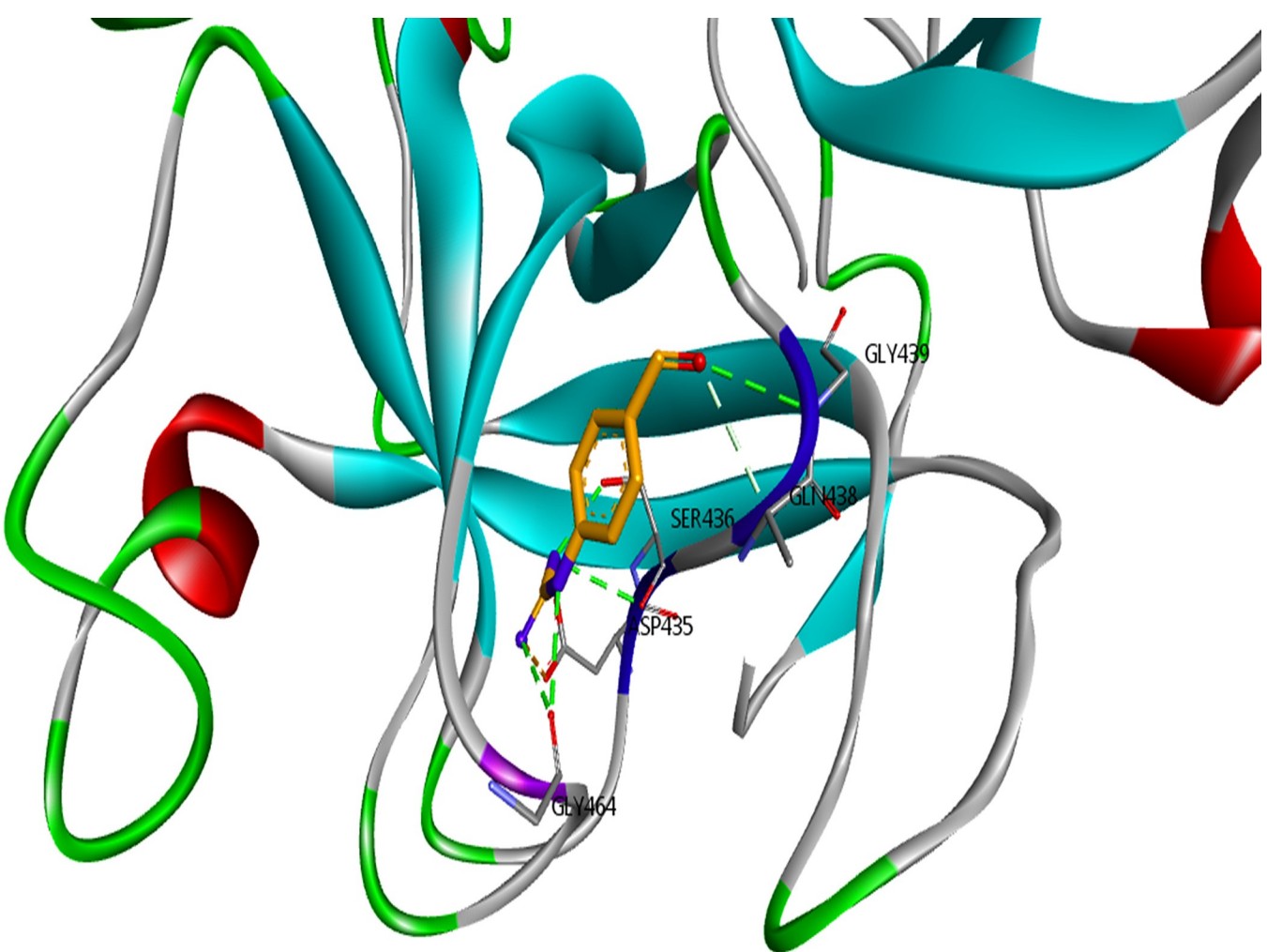

**Fig 6. Binding interactions of Nafamostat as reference inhibitor of TMPRSS2.** The hydrogen bonds are shown in green dotted lines and residues with van der Waals interactions placed around the ligand with white dotted lines.

GLN189, ASP 216, ARG222, as well as ASN274 that do not locate in the binding site, allocated the high fluctuations during the simulation of all systems. All amino acids interacted with $L_6$ in molecular docking except GLN189 demonstrated low values of fluctuation, which confirms the stability of interactions with Mpro without conformation changes. Residues ASN142 and ASP187, residues THR190 and GLN192 belong to $L_7$-Mpro and $L_3$-Mpro complex interactions in molecular docking studies which more fluctuated than other amino acids involved in ligand binding. These results affirm less conformational deviation and instability of $L_6$-Mpro complex than $L_7$-Mpro and $L_3$- Mpro complexes.

Fig 9B shows the fluctuations of TMPRSS2 residues and its complex with three ligands ($L_3$, $L_6$, and $L_7$) during the period. The highly flexible regions of the protein are residues 200–210 and 246–264 and the active site residues remained stable with low deviation during simulation. $L_6$-TMPRSS2 complex plot demonstrated the best alignment with TMPRSS2 graph, whereas $L_3$-TMPRSS2 complex plot was less coincidental with the protein fluctuations. On the other hand, The $L_7$ ligand bound to the TMPRSS2 protein generally has higher fluctuations than $L_6$ and $L_3$ ligands. Most of these changes are in the $S_1$ subdomain, and these data indicated the instability of the TMPRSS2 protein in binding to $L_7$, which is confirmed by RMSD analysis.

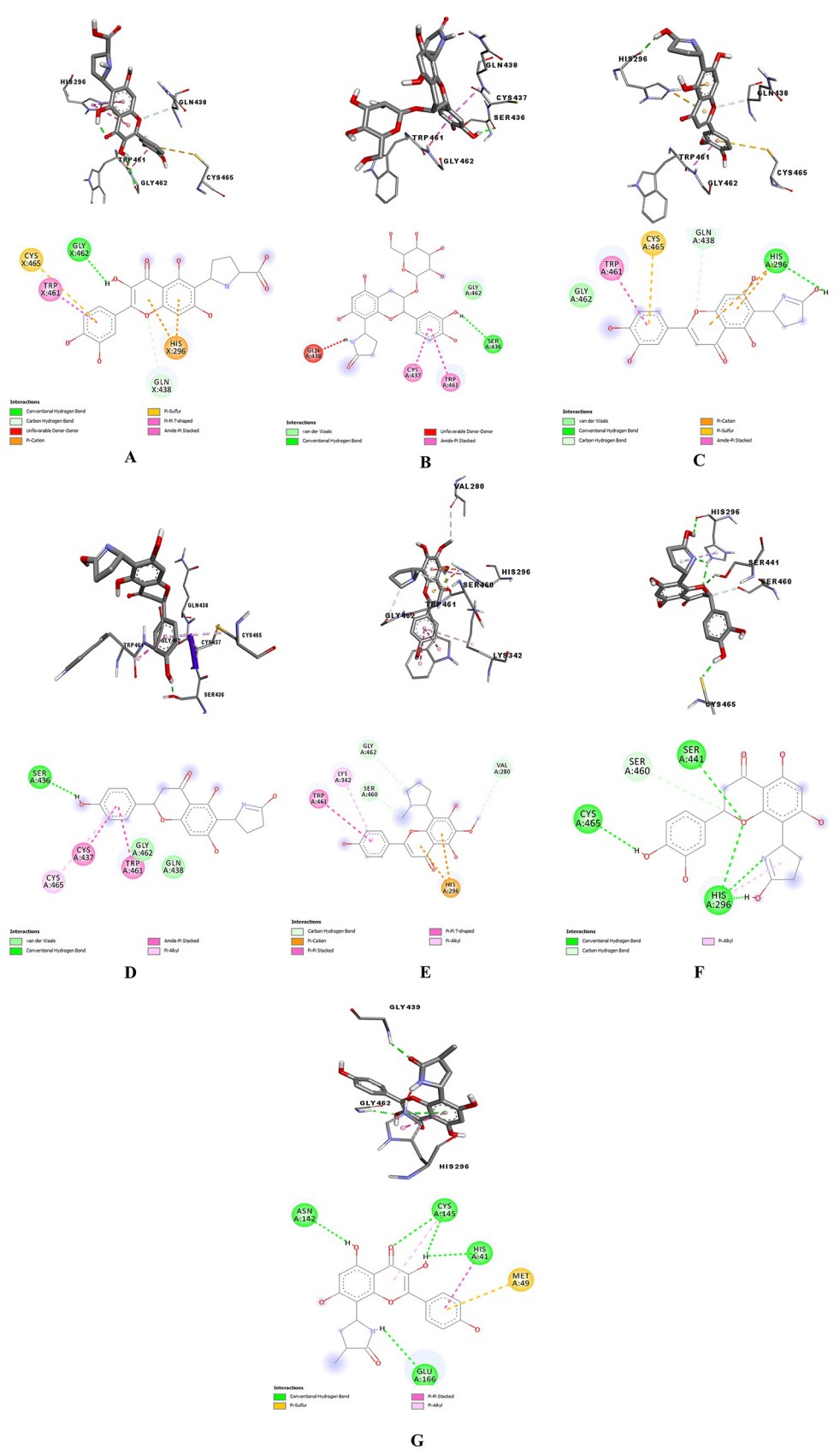

**Fig 7.** 3D (up) and 2D (down) binding interactions of $L_1$ (A), $L_2$ (B), $L_3$ (C), $L_4$ (D), $L_5$ (E), $L_6$ (F), and $L_7$ (G) with TMPRSS2.

It is now necessary to assess the magnitude of fluctuations caused by the binding of the ligands to the target proteins. A very high increase in the mobility of the roots can lead to an increase in the size of protein 3-D structure. The possible change in dimension and compactness of the protein was examined using the radius of gyration (Rg) and solvent-accessible surface area (SASA) analyses.

The Rg of the protein indicates the degree of compaction of the protein and a high increase in Rg values indicates instability in the system [44, 53]. As shown in Fig 10, the Rg value remained almost constant for all systems; however, the conformation of Mpro was affected by $L_3$, $L_6$, and $L_7$ due to their slight increase in Rg value (Fig 10A). Moreover, the most exposed conformation of Mpro was with $L_6$ ligand because of low variations in the compactness of protein. As shown in Fig 10B, except for the $L_7$ ligand attached to TMPRSS2, by ignoring minor changes during the simulation for $L_3$ and $L_6$ ligands, it can be concluded that the ligand systems are stable. In addition, the binding of the $L_7$ ligand to TMPRSS2 caused significant changes in the dimensions of the protein. These changes showed consistent with RMSD plot (Fig 8B), and the increased vibrations of protein residues and Rg values indicated that the $L_7$-bound TMPRSS2 protein was very unstable. These results correspond to those of RMSF with the most fluctuations because more movements in the roots can increase the protein radius and decrease its compactness.

To evaluate changes in the protein 3-D structure and its stability, SASA value was computed, which represents the solvent accessibility of the protein. This analysis predicts the number of protein surface amino acids that can be accessible to the solvent [54, 55]. A higher SASA value indicates protein unfolding reactions. As shown in Fig 11A and 11B, the least fluctuations occurred in all systems during the simulation. This analysis shows that the folding of all studied proteins has remained almost constant. The results of the TMPRSS2-bound $L_7$ ligand (Fig 11B) show that the solvent-accessible surface did not change suddenly in this protein. Therefore, the increase in the Rg value of this structure was analyzed using its structural information before and after the peak in the previous graphs. The structural alignment results of

**Table 3. Molecular binding results of docked compounds with TMPRSS2.**

| TMPRSS2 (7MEQ) | Hydrogen bond | Distance (Å) | Donor Angle (degree) | Other Interaction |
|---|---|---|---|---|
| $L_1$ | GLY462 | 3.07 | 118.62 | HIS296, GLN438, CYS465, TRP461 |
| $L_2$ | SER436 | 2.30 | 168.16 | GLY462, TRP461, CYS437 |
| $L_3$ | HIS296 | 2.05 | 159.30 | GLU299, GLN438 |
| | GLN438 | 2.22 | 131.29 | |
| $L_4$ | SER436 | 1.84 | 163.95 | GLN438, CYS437, GLY462, CYS465 TRP461 |
| $L_5$ | - | - | - | HIS296, VAL280, GLY462, SER460, LYS342, TRP461 |
| $L_6$ | HIS296 | 2.03 | 142.27 | SER460, HIS296 |
| | HIS296 | 2.98 | 130.35 | |
| | HIS296 | 2.78 | 117.15 | |
| | SER441 | 2.28 | 144.02 | |
| | CYS465 | 2.59 | 127.04 | |
| $L_7$ | HS296 | 2.12 | 153.80 | GLN438, THR459 |
| | GLY439 | 2.17 | 139.10 | |
| | GLY462 | 2.29 | 160.09 | |

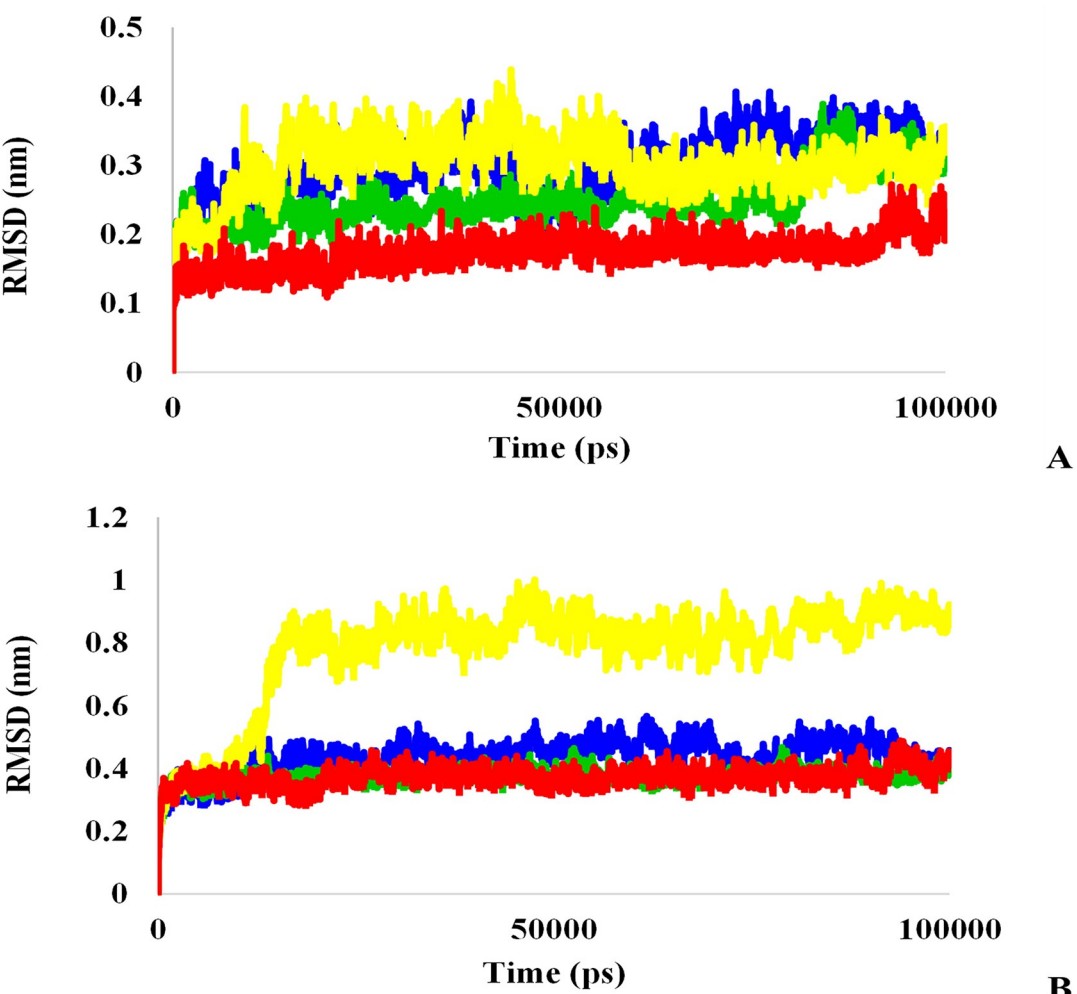

**Fig 8.** A: RMSD analysis of Mpro protein (red), $L_3$-Mpro (blue), $L_6$-Mpro (green), and $L_7$-Mpro (yellow), B: RMSD analysis of TMPRSS2 protein (red), $L_3$-TMPRSS2 (blue), $L_6$-TMPRSS2 (green), and $L_7$-TMPRSS2 (yellow).

the protein show that the $L_7$ influences the active site and domain I of this protein, and leads to protein instability due to structural changes (Fig 12).

Since hydrogen bonds play an essential role in the binding strength of ligands to proteins, hydrogen bonds were investigated for all complexes between ligands and proteins during the simulation. Figs 13 and 14 illustrate the completely different behaviors of hydrogen bonds in these three ligands that the number of hydrogen bonds between tested ligands and the proteins in the docking stage reduced after MD simulation. $L_6$ formed the utmost hydrogen bonds (3 HBonds) with Mpro protein while $L_3$ and $L_7$ were able to form 2 and 1 hydrogen bonds on average, respectively. On the other hand, all ligands bound to TMPRSS2 protein with approximately 2 hydrogen bonds. Although $L_6$ had more potential to form hydrogen bonds with the receptor because of its lowest binding free energy, these bonds are not stable and the number of hydrogen bonds decreased within 50 ns suggesting the key role of hydrophobic bonds in the inhibition of TMPRSS2.

Further investigation for a better understanding of the behavior of proteins when binding to ligands is Principal Component Analysis (PCA) because it results in the reduction of the high dimensionality of massive related data including atom coordinates and dihedral angles

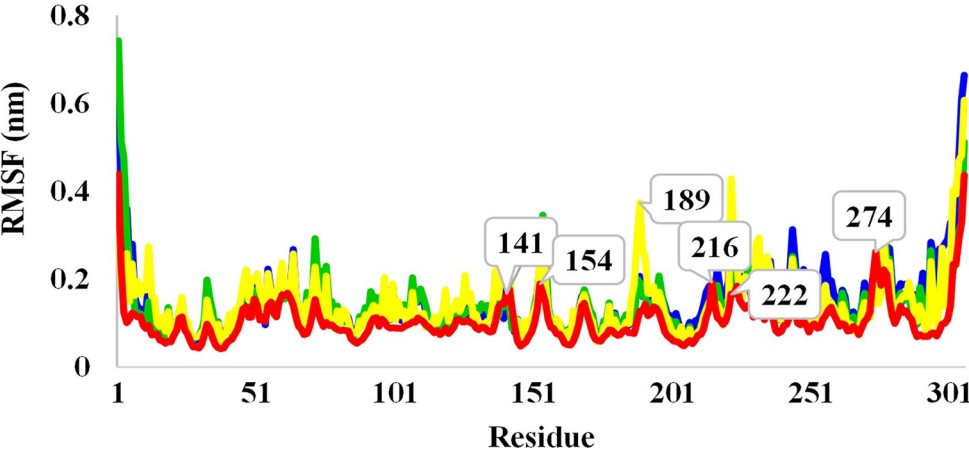

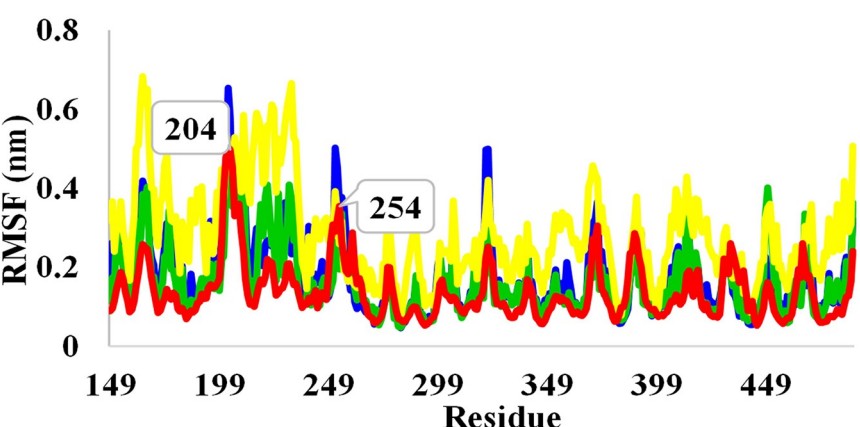

**Fig 9.** A: RMSF analysis of Mpro protein (red), L₃-Mpro (blue), L₆-Mpro (green), and L₇-Mpro (yellow), B: RMSF analysis of TMPRSS2 protein (red), L₃-TMPRSS2 (blue), L₆-TMPRSS2 (green), and L₇-TMPRSS2 (yellow).

over the entire period of simulation. Fig 15 displays the reflection of the first and second principal components of the Mpro enzyme. As shown in this figure, Mpro behaved differently in the vector coordinates when bound to ligands. In addition, L₆ and L₇ ligands (Fig 15B and 15C) occupied less space than Mpro (Fig 15D), which shows their compact structure. The diagrams of these two ligands contain specific clusters that indicate the local movements of the protein when binding to these two compounds. The less-occupied space of L₆ indicated that the protein domain movement was limited during binding and therefore it occupies less space. Moreover, the L₆ plot shifted to the positive amounts of PC1 in comparison with Mpro pattern, whereas the maintenance of L₃ and L₇ position can be seen along the PC1 axis in Fig 15A and 15C. This declares a global translation of L₆-Mpro complex to the right. Therefore, the stability of this inhibitor with Mpro following the right-oriented movements of the protein can be the result, since L₆ revealed the most negative docking score and stable complex structure with Mpro.

The distribution of points can describe the conformational changes or transitions over the period. In this regard, the separated points of L₃ plot (Fig 15A) compared to Mpro suggest the conformational changes induction in Mpro along both PCs. Fig 16 shows the first and second

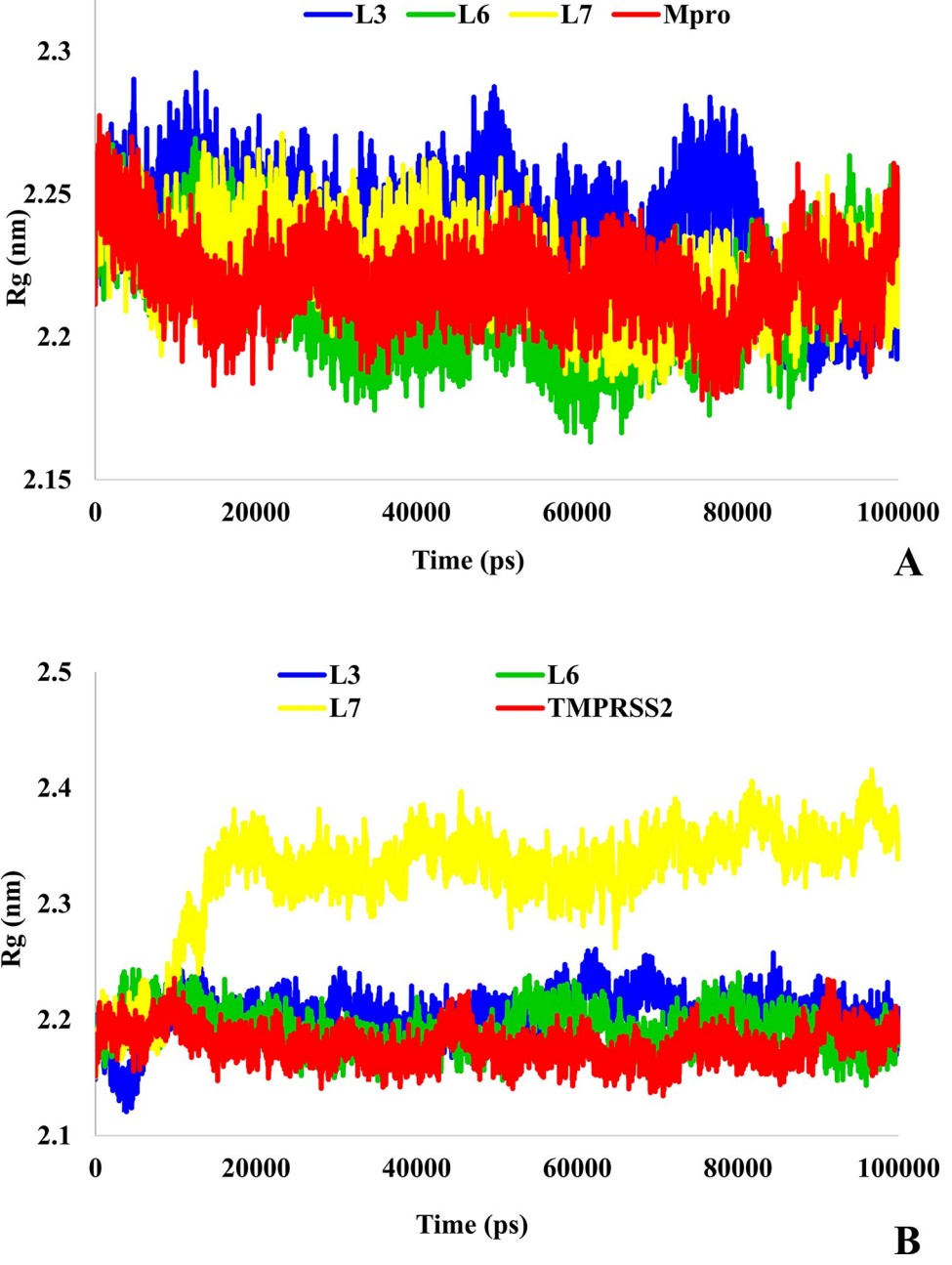

**Fig 10.** A: Rg analysis of Mpro protein (red), $L_3$-Mpro (blue), $L_6$-Mpro (green), and $L_7$-Mpro (yellow), B: Rg analysis of TMPRSS2 protein (red), $L_3$-TMPRSS2 (blue), $L_6$-TMPRSS2 (green), and $L_7$-TMPRSS2(yellow).

principal component reflections for the TMPRSS2 enzyme, presenting the structural changes of the protein upon binding to all three ligands. Although there are differences in the graphs, $L_7$ and $L_3$ have induced less compression in the protein by binding to it (Fig 16A and 16C). The enzyme experienced fewer changes in its conformation when binding to L6 (Fig 16D). As it is clear in Fig 16C, Mpro underwent the most conformational changes when interacting with $L_7$ ligand according to the sporadic distribution of $L_7$ plot points. These alternations are

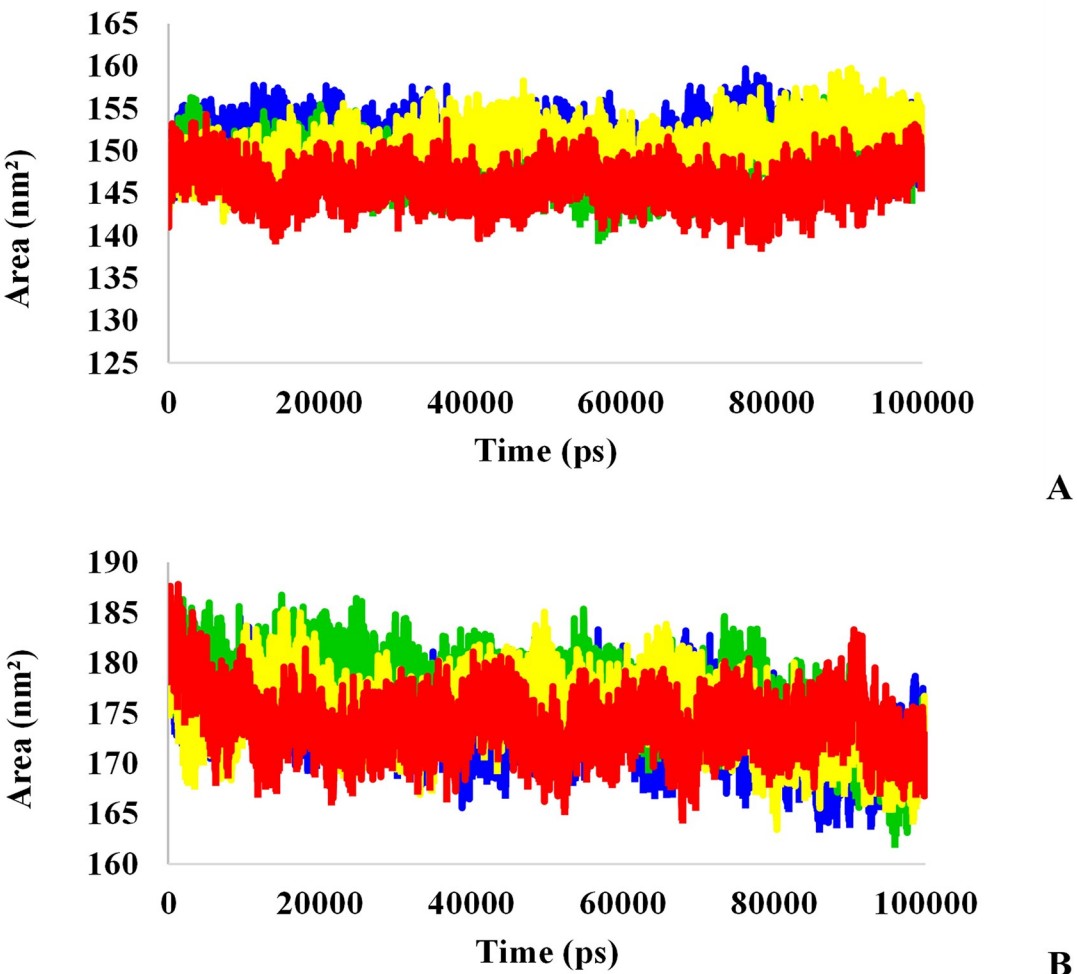

**Fig 11.** A: SASA analysis of Mpro protein (red), $L_3$-Mpro (blue), $L_6$-Mpro (green), and $L_7$-Mpro (yellow), B: SASA analysis of TMPRSS2 protein (red), $L_3$-TMPRSS2 (blue), $L_6$-TMPRSS2 (green), and $L_7$-TMPRSS2(yellow).

consistent with increased Rg and SASA values of TMPRSS2 induced by $L_7$. As a result, $L_7$ leads to the instability and structural changes of the protein.

Finally, after the complete equilibrium of the complexes, the binding free energies were calculated during the last 200 frames of 20 ns molecular dynamics simulation by Poisson-Boltzmann (MM-PBSA) to illuminate the most potent ligand. The obtained results are presented in Figs 17 and 18. The very low values of the binding energies confirmed the stability of the complexes and proper binding of ligands into protein active sites of proteins. The results indicate that van der Waals forces play a very effective role in binding ligands to proteins. The most negative van der Waals energy belongs to $L_6$ that declares the highest shape complementarity between $L_6$ and Mpro compared to the other ligands and indicates consistency with the results of docking stage. Moreover, electrostatic energy as an indicator of salt bridges stability and hydrogen bonds between the inhibitor and the enzyme, shows Coulombic interactions between charge groups. Based on MM-PBSA results, the most favorable electrostatic energy is for $L_6$ which confirms hydrogen bonds analysis results (Fig 13B). Although the total binding energy of L6 is lower than that of L7, it has the most inhibition potential of Mpro due to the

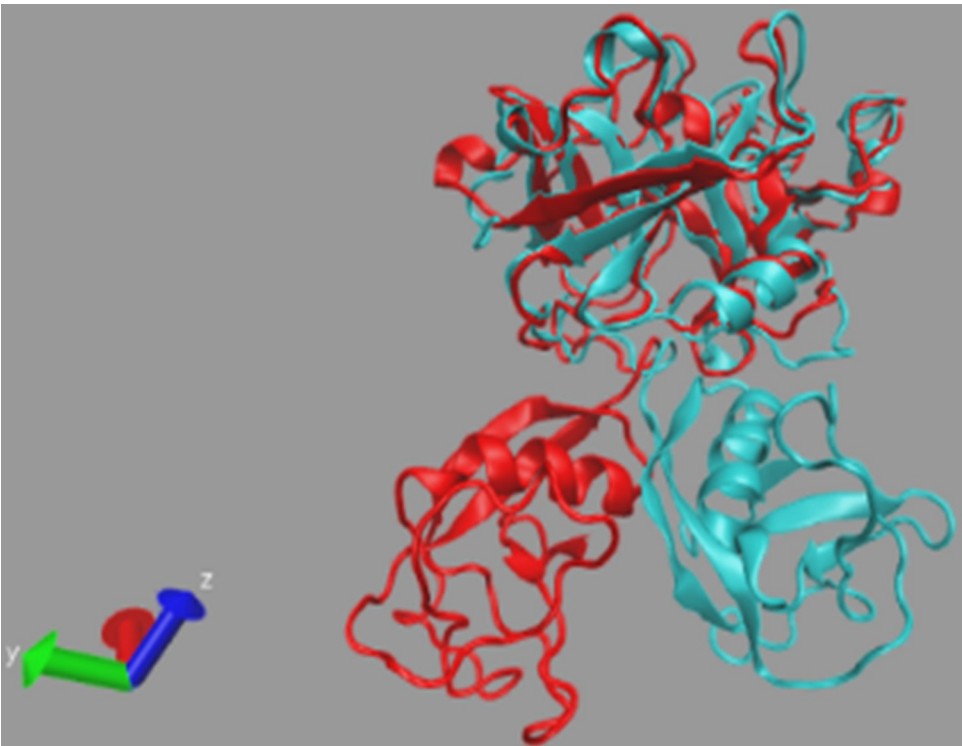

**Fig 12. Alignment analysis of TMPRSS2 protein tertiary structure.**

most hydrogen bond interactions. Also, Polar solvation energy is positive for all the tested ligands.

For the MM-PBSA results of ligands-TMPRSS2 complexes in Fig 18, $L_3$ had higher van der Waals and total binding energy than $L_6$ and $L_7$; however, the most negative electrostatic energy obtained for $L_6$ that elucidates its potential hydrogen bonds and its more inhibitory effect on TMPRSS2.

## Conclusions

In this research, seven alkaloid ligands with quercetin nuclei were used to interact with the main protease active site of coronavirus and the TMPRSS2 protein as the host of coronavirus. First, three ligands were selected from the seven alkaloid ligands using molecular docking. In the present study, the inhibition mechanism of the Mpro of COVID-19 and the host receptor of TMPRSS2 with three ligands Dracocephin-A ($L_3$), Phyllospadine ($L_6$) and Prolin-A ($L_7$) were studied step by step using molecular dynamics simulation in 100 ns. RMSD results showed that all systems were completed equilibrium after 20 ns. The analysis of structural parameters RMSF, Rg, SASA, PCA, hydrogen bonds, and binding energy for Mpro, TMPRSS2, and $L_3$, $L_6$, and $L_7$ complexes with Mpro and the cellular receptor showed that Mpro had a stable structure with high strength. The $L_3$, $L_6$, and $L_7$ were fitted in the binding site of Mpro. His41 amino acid of domain I, as the active site of Mpro, is completely covered by all three inhibitors $L_3$, $L_6$, and $L_7$ and becomes inaccessible. The results also reveal that inhibitors can form stable hydrogen bonds with amino acids, which helps the binding strength of inhibitors in the binding site. In total, the high binding energy between L3, L6, and L7 inhibitors and the binding site, as well as the complete coverage of His41 amino acid of Mpro by these inhibitors

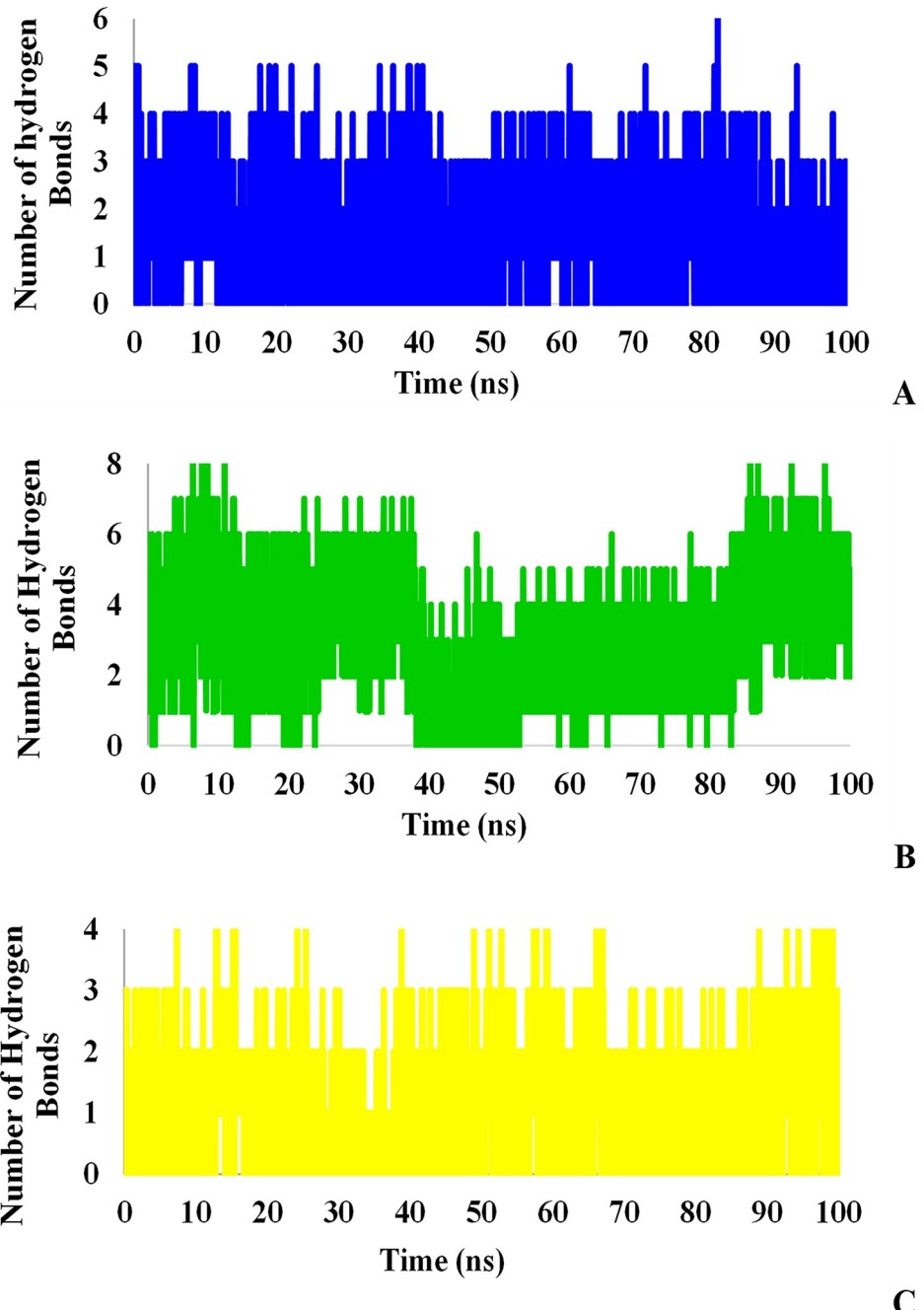

**Fig 13.** Hydrogen bonds between Mpro and $L_3$ (A), $L_6$ (B), and $L_7$ (C).

indicated that these three inhibitors are suitable candidates as drugs to inhibit the enzyme activity of coronavirus-SARS-2. As for TMPRSS2, the residues His 296, Ser 441, and Gly 462 of the $S_1$ domain are known as the active site of the TMPRSS2 enzyme. These three sites are covered by all three inhibitors $L_3$, $L_6$, and $L_7$, and become inaccessible; however, the enzyme was unstable in binding to the Prolin-A because this ligand changed the third structure of the protein in the $S_1$ domain. According to the results, the inhibitors can form several hydrogen

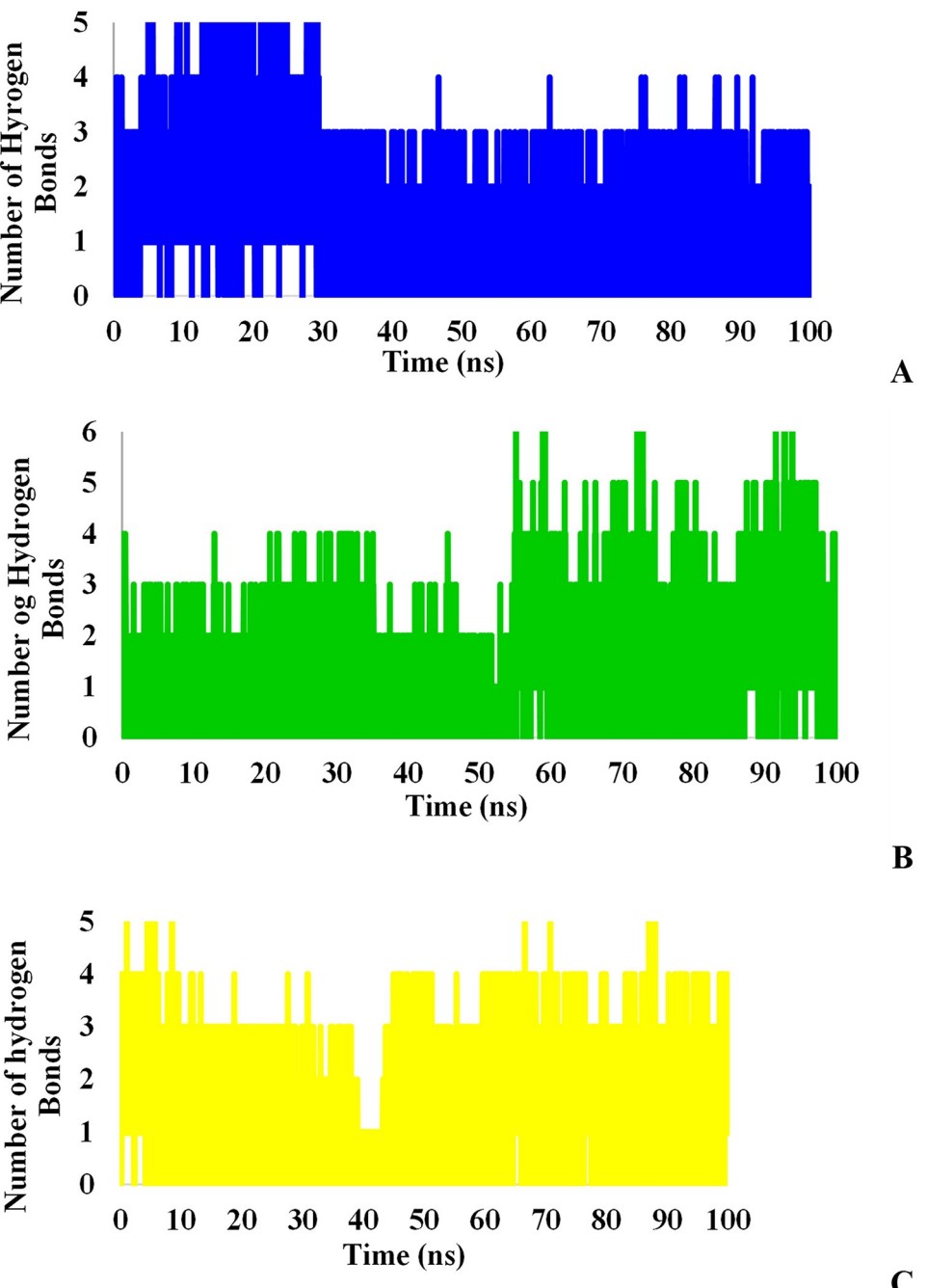

**Fig 14.** Hydrogen bonds between TMPRSS2 and L$_3$ (A), L$_6$ (B), and L$_7$ (C).

bonds with some amino acids, which helps the binding strength of the inhibitors in the binding site. In total, the high binding energy between L$_3$ and L$_6$ inhibitors and the binding site as well as the complete coverage of His 296, Ser 441, and Gly 462 amino acids of the TMPRSS2 protease by these inhibitors indicate that these two inhibitors are appropriate candidates as drugs to inhibit TMPRSS2 enzyme activity.

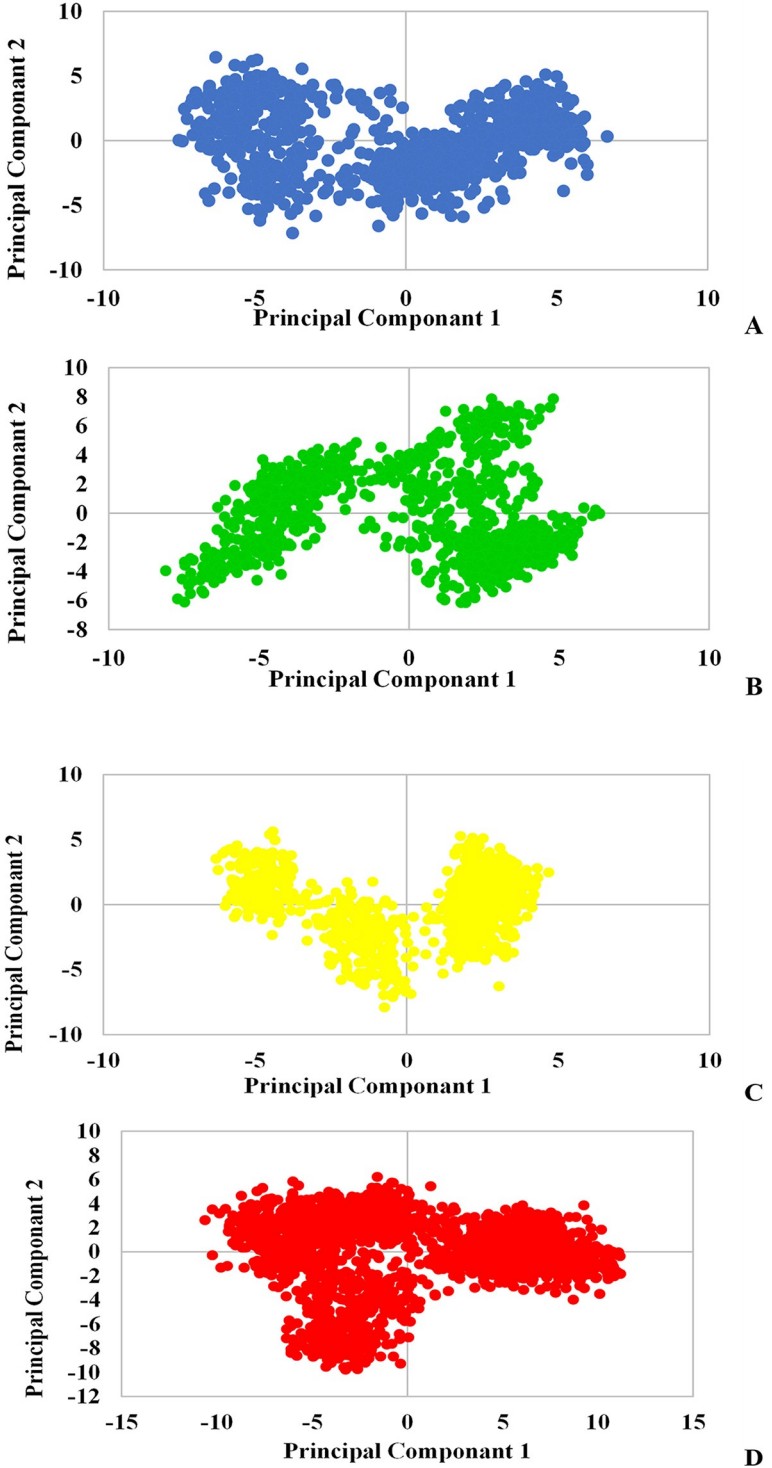

**Fig 15.** PCA plots for $L_3$ (A), $L_6$ (B), $L_7$ (C) and Mpro (D): following Conformational Space and Dynamics during the period of simulations.

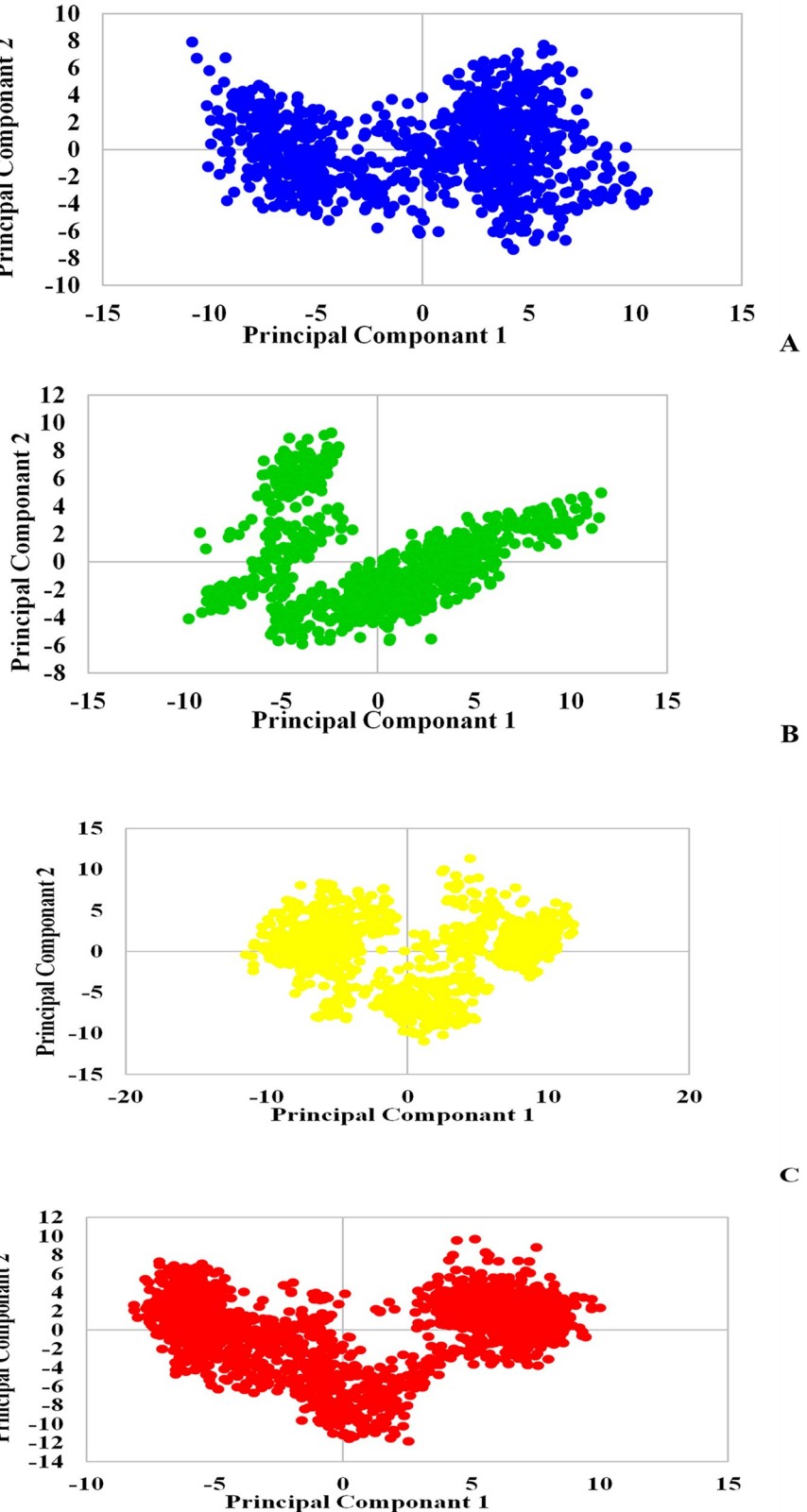

**Fig 16.** PCA plots for $L_3$ (A), $L_6$ (B), $L_7$ (C) and TMPRSS2 (D): following Conformational Space and Dynamics during the period of simulations.

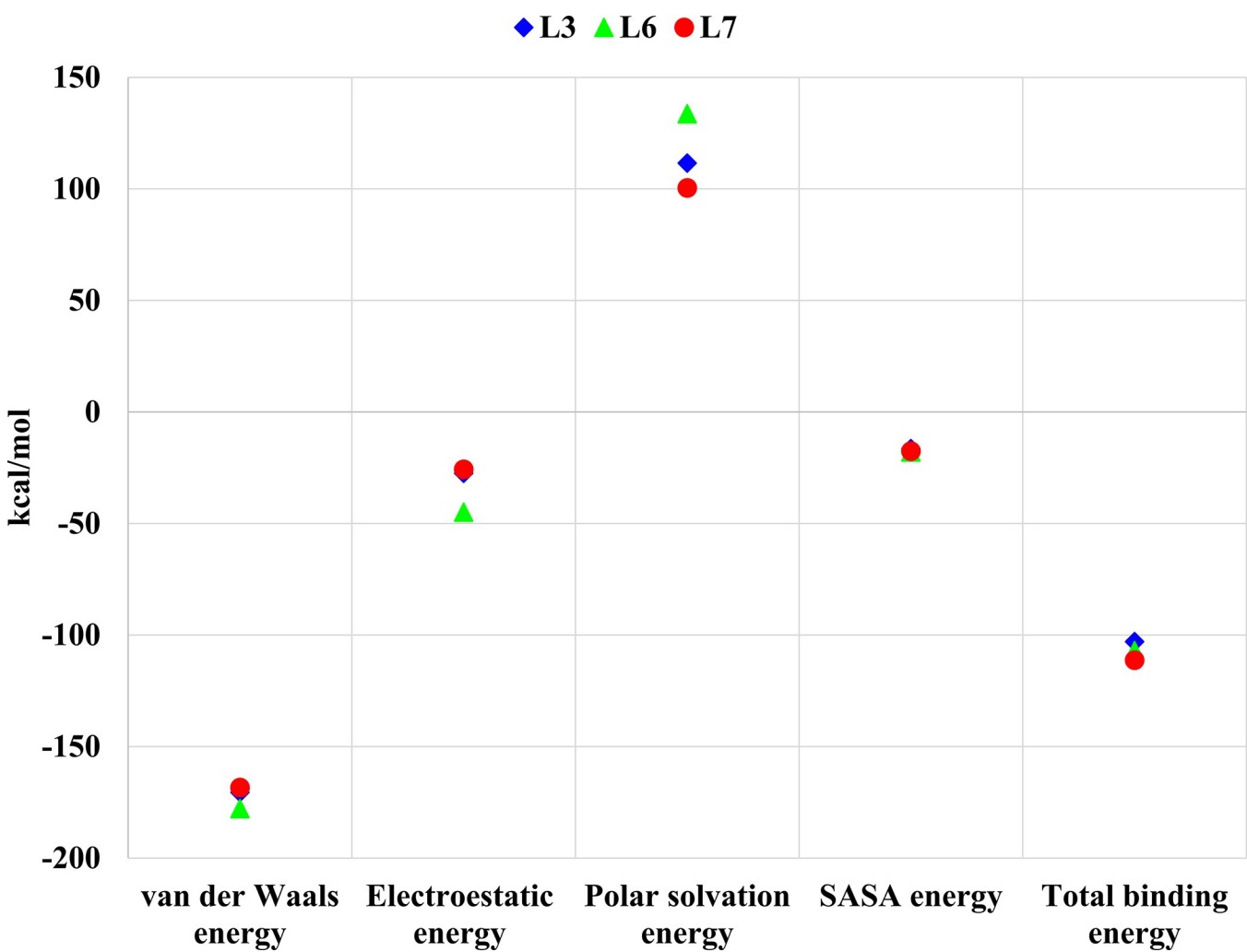

**Fig 17. Binding free energies calculation of Mpro-$L_3$, Mpro-$L_6$, and Mpro-$L_7$.**

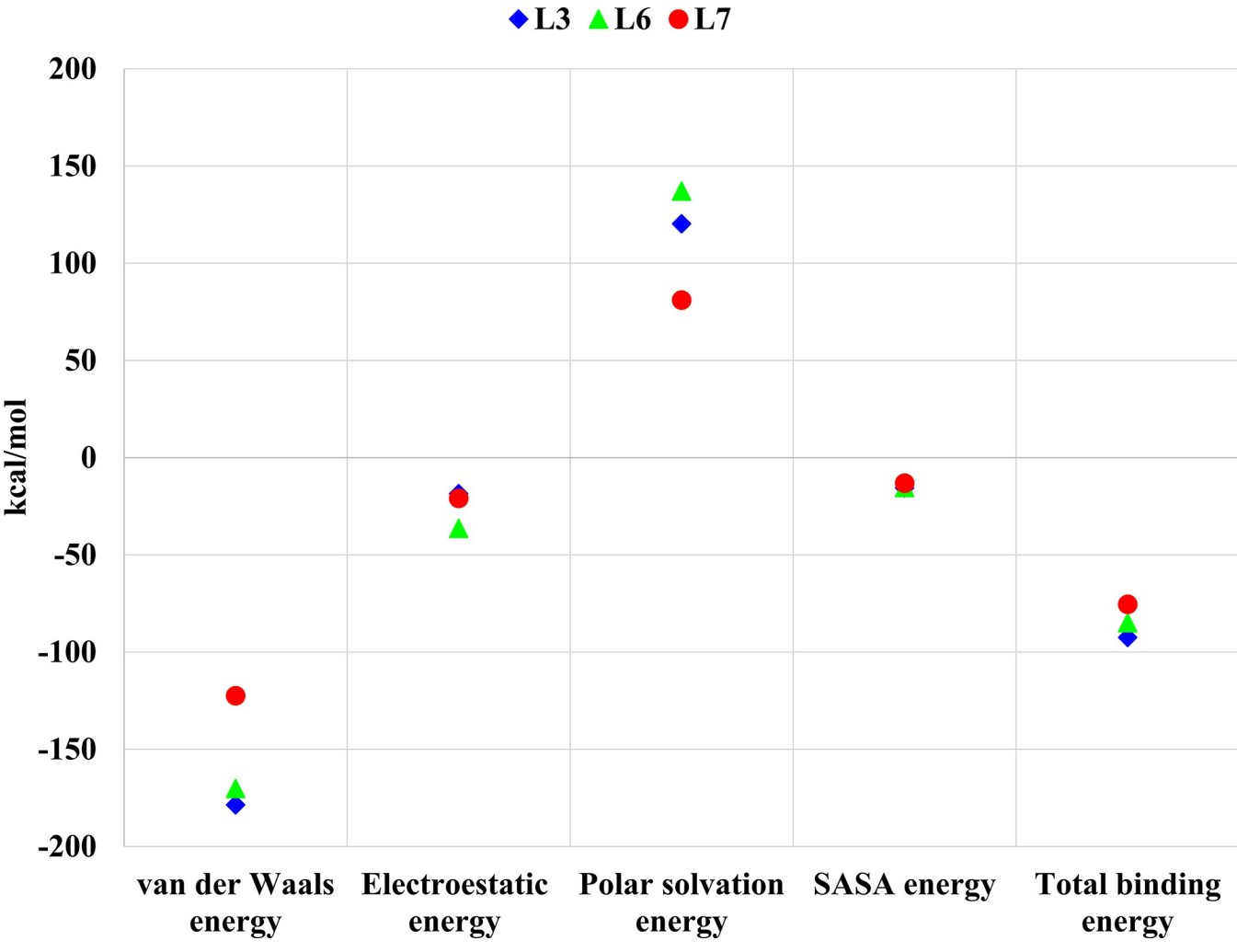

**Fig 18. Binding free energies calculation of TMPRSS2-L$_3$, TMPRSS2-L$_6$, and TMPRSS2-L$_7$.**

## Acknowledgments

The authors would like to express their appreciation to the management of computer center of the Chemical Engineering Department, Shahid Bahonar University of Kerman, Kerman, Iran for supporting this work.

## Author Contributions

**Conceptualization:** Ali Mohebbi, Marzieh Eskandarzadeh, Marzie Fatehi.

**Formal analysis:** Marzieh Eskandarzadeh.

**Investigation:** Hanieh Zangi.

**Methodology:** Ali Mohebbi, Hanieh Zangi, Marzie Fatehi.

**Software:** Hanieh Zangi.

**Supervision:** Ali Mohebbi, Marzie Fatehi.

**Validation:** Hanieh Zangi.

**Writing – original draft:** Marzieh Eskandarzadeh.

**Writing – review & editing:** Ali Mohebbi, Marzie Fatehi.

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
