## [Decision Letter · Decision Letter 0]

25 Oct 2023

PONE-D-23-31183In silico study of alkaloids with quercetin nucleus for inhibition of SARS-CoV-2 protease and receptor cell proteasePLOS ONE

Dear Dr. Mohebbi,

Thank you for submitting your manuscript to PLOS ONE. After careful consideration, we feel that it has merit but does not fully meet PLOS ONE’s publication criteria as it currently stands. Therefore, we invite you to submit a revised version of the manuscript that addresses the points raised during the review process.

We look forward to receiving your revised manuscript.

Kind regards,

Ahmed A. Al-Karmalawy, Ph.D.

Academic Editor

PLOS ONE

Journal Requirements:

Reviewers' comments:

Reviewer's Responses to Questions

**Comments to the Author**

1. Is the manuscript technically sound, and do the data support the conclusions?

Reviewer #1: Yes

Reviewer #2: Yes

2. Has the statistical analysis been performed appropriately and rigorously? 

Reviewer #1: Yes

Reviewer #2: Yes

3. Have the authors made all data underlying the findings in their manuscript fully available?

Reviewer #1: Yes

Reviewer #2: Yes

4. Is the manuscript presented in an intelligible fashion and written in standard English?

Reviewer #1: Yes

Reviewer #2: Yes

5. Review Comments to the Author

Reviewer #1: COMMENTS TO AUTHOR:

This manuscript presents an in silico investigation involving the study of seven alkaloid ligands containing a quercetin nucleus for their potential to inhibit the SARS-CoV-2 protease and receptor cell protease, namely Mpro and TMPRSS2. The study works on computational drug design techniques, including molecular docking and molecular dynamics simulations. The article exhibits good writing and organization, but there are some concerns that need to be addressed before it can be considered suitable for publication.

1. In the abstract, in line ‘Covid-19 disease because the disease has killed so many people…..In place of a killed word, the author can use another word.

2. Throughout the manuscript, please ensure that the term "In silico" is presented in italic font.

3. What is the specific reason for selecting PDB entries 6LU7 and 7MEQ as the targets for the molecular docking study?

4. For molecular docking, in the study in Fig. 2 and 3, the author should provide a 3D binding interactions diagram of standards (Camostat ) that they used.

5. Figure 9A incorrectly labels the graph scale as "Time (nm)," but it should be correctly labeled as "Time (ns)." Please make this correction for accuracy or verified ones.

6. Three ligands, Dracocephin-A (L₃), Phyllospadine (L₆), and Prolin-A (L₇), were studied step by step using molecular dynamics simulation in 100 ns. Did the author perform molecular dynamics simulation with different 50 ns?

7. What common functional groups or structural characteristics in all seven alkaloid ligands are responsible for achieving favorable molecular docking scores and establishing significant intermolecular interactions with the selected targets?

8. In reference 14, the Journal name ‘cell ’ C should be capitalized. ‘Cell ’

Reviewer #2: The authors provided a computational approach for exploring the anti-SARS-CoV-2 activity of quercetin-containing alkaloids with potentiality for Mpro inhibition. Authors adopted molecular docking-coupled dynamics simulation which highlighted the role of preferential residue-wise interactions and dynamic flexibility of the protein structure on ligand’s binding/affinity profiles. This manuscript is relevant and valuable in the field, with some suggestions prior publication.

1. Authors provide comprehensive residue-wise interaction with the simulated ligands at Mpro target, however brief introduction for the target topology and key secondary structures and motifs should be presented and augmented with 3D representation of the whole target highlighting them.

2. Authors highlighted polar hydrogen bonds and hydrophobic contacts. However, hydrogen binding should be presented within hydrogen bond distances as well as bond angles since hydrogen bond depend on both. Authors should mention the Hydrogen bond angles as well as their distances, since the strength of hydrogen bonding is based on both parameters in a way to ensure the adequacy of optimum hydrogen bonding.

3. Tables 4 and 5, better represented as figures for better differential free binding energy demonstration and visualization.

4. Authors should elaborate more on the discussion section through presenting comparative findings from reported literature studies that investigated reported and/or close compounds against the same target protein.

6. PLOS authors have the option to publish the peer review history of their article (what does this mean?). If published, this will include your full peer review and any attached files.

Reviewer #1: No

Reviewer #2: **Yes**

---

## [Author Response · Author response to Decision Letter 0]

11 Jan 2024

Reviewers Comments Response

Title: “In silico study of alkaloids with quercetin nucleus for inhibition of SARS-CoV-2 protease and receptor cell protease”

Ali Mohebbi, Marzieh Eskandarzadeh, Hanieh Zangi, Marzie Fatehi

The authors would like to thank the reviewers for their comprehensive and constructive report that helped us improve the quality of our manuscript.

Comments from the editors and reviewers:

-Reviewer 1 (Changes in response to Reviewer 1 were highlighted with turquoise colour)

This manuscript presents an in silico investigation involving the study of seven alkaloid ligands containing a quercetin nucleus for their potential to inhibit the SARS-CoV-2 protease and receptor cell protease, namely Mpro and TMPRSS2. The study works on computational drug design techniques, including molecular docking and molecular dynamics simulations. The article exhibits good writing and organization, but there are some concerns that need to be addressed before it can be considered suitable for publication.

1. In the abstract, in line ‘Covid-19 disease because the disease has killed so many people….’. In place of a killed word, the author can use another word. 

Response: The sentence was changed to “Researchers are studying the design and discovery of drugs to inhibit the SARS-CoV-2 virus due to its high mortality rate.”. 

2. Throughout the manuscript, please ensure that the term "In silico" is presented in italic font. 

Response: Its font was changed to italic throughout the manuscript. 

3. What is the specific reason for selecting PDB entries 6LU7 and 7MEQ as the targets for the molecular docking study? 

Response: The 6LU7 is the main protease (Mpro) of SARS-CoV-2 and a key enzyme of coronaviruses, which has a pivotal role in mediating viral replication and transcription. Therefore, 6LU7 can be considered as drug target for SARS-CoV-2 (https://doi.org/10.1038/s41586-020-2223-y ). On the other hand, the 7MEQ is a key host cell factor for viral entry and pathogenesis of SARS-CoV-2 and a broad-spectrum synthetic serine protease inhibitor (https://doi.org/10.1038/s41589-022-01059-7 ). 

4. For molecular docking, in the study in Fig. 2 and 3, the author should provide a 3D binding interactions diagram of standards (Camostat) that they used. 

Response: We provided the 3D binding interactions of N3 as the reference inhibitor of Mpro and Nafamostat as the reference inhibitor of TMPRSS2 in 7MEQ code. Camostat is another reference inhibitor of TMPRSS2 that we compared its structure with L₆. Also, the manuscript was promoted: 

• The second paragraph in the section “Molecular docking studies of Mpro and TMPRSS2 proteins in complex with 7 alkaloid ligands”. 

5. Figure 9A incorrectly labels the graph scale as "Time (nm)," but it should be correctly labeled as "Time (ns)." Please make this correction for accuracy or verified ones. 

Response: The unit of time was corrected in Figure 9A. 

6. Three ligands, Dracocephin-A (L₃), Phyllospadine (L₆), and Prolin-A (L₇), were studied step by step using molecular dynamics simulation in 100 ns. Did the author perform molecular dynamics simulation with different 50 ns? 

Response: The MD simulations were performed only for 100 ns.

7. What common functional groups or structural characteristics in all seven alkaloid ligands are responsible for achieving favorable molecular docking scores and establishing significant intermolecular interactions with the selected targets? 

Response: The flavone structure is responsible for ligand binding to Mpro and the more hydrogen interactions formed by the pyrrolidine structure with TMPRSS2 catalytic site. 

8. In reference 14, the Journal name ‘cell’ C should be capitalized. ‘Cell’. 

Response: The journal name was changed to “Cells”. 

Reviewer 2 (Changes in response to Reviewer 2 were highlighted with bright green colour)

The authors provided a computational approach for exploring the anti-SARS-CoV-2 activity of quercetin-containing alkaloids with potentiality for Mpro inhibition. Authors adopted molecular docking-coupled dynamics simulation which highlighted the role of preferential residue-wise interactions and dynamic flexibility of the protein structure on ligand’s binding/affinity profiles. This manuscript is relevant and valuable in the field, with some suggestions prior publication. 

1. Authors provide comprehensive residue-wise interaction with the simulated ligands at Mpro target, however brief introduction for the target topology and key secondary structures and motifs should be presented and augmented with 3D representation of the whole target highlighting them. 

Response: We provided the key regions of proteins structures in 3D view with a short description in the text of manuscript (The first paragraph in the section “Molecular docking studies of Mpro and TMPRSS2 proteins in complex with 7 alkaloid ligands”.). 

2. Authors highlighted polar hydrogen bonds and hydrophobic contacts. However, hydrogen binding should be presented within hydrogen bond distances as well as bond angles since hydrogen bond depend on both. Authors should mention the Hydrogen bond angles as well as their distances, since the strength of hydrogen bonding is based on both parameters in a way to ensure the adequacy of optimum hydrogen bonding. 

Response: The distance and donor angle of conventional hydrogen bonds were given in Tables 2 and 3.

3. Tables 4 and 5, better represented as figures for better differential free binding energy demonstration and visualization. 

Response: The binding energy data were shown as Figs 17 and 18. These tables were removed from the manuscript.

4. Authors should elaborate more on the discussion section through presenting comparative findings from reported literature studies that investigated reported and/or close compounds against the same target protein. 

Response: We discussed the results and discussion more in the molecular docking section. For example: 

• The second paragraph in the section “Molecular docking studies of Mpro and TMPRSS2 proteins in complex with 7 alkaloid ligands”. 

• The third paragraph in the section “Molecular docking studies of Mpro and TMPRSS2 proteins in complex with 7 alkaloid ligands”.

---

## [Decision Letter · Decision Letter 1]

22 Jan 2024

In silico study of alkaloids with quercetin nucleus for inhibition of SARS-CoV-2 protease and receptor cell protease

PONE-D-23-31183R1

Dear Dr. Mohebbi,

We’re pleased to inform you that your manuscript has been judged scientifically suitable for publication and will be formally accepted for publication once it meets all outstanding technical requirements.

Kind regards,

Ahmed A. Al-Karmalawy, Ph.D.

Academic Editor

PLOS ONE

Reviewers' comments:

Reviewer's Responses to Questions

**Comments to the Author**

1. If the authors have adequately addressed your comments raised in a previous round of review and you feel that this manuscript is now acceptable for publication, you may indicate that here to bypass the “Comments to the Author” section, enter your conflict of interest statement in the “Confidential to Editor” section, and submit your "Accept" recommendation.

Reviewer #1: All comments have been addressed

Reviewer #2: (No Response)

2. Is the manuscript technically sound, and do the data support the conclusions?

Reviewer #1: Yes

Reviewer #2: (No Response)

3. Has the statistical analysis been performed appropriately and rigorously? 

Reviewer #1: N/A

Reviewer #2: (No Response)

4. Have the authors made all data underlying the findings in their manuscript fully available?

Reviewer #1: Yes

Reviewer #2: (No Response)

5. Is the manuscript presented in an intelligible fashion and written in standard English?

Reviewer #1: Yes

Reviewer #2: (No Response)

6. Review Comments to the Author

Reviewer #1: The revised version of the manuscript is now acceptable for publication, as the author has incorporated all comments and suggestions.

Reviewer #2: Authors adequately responded to comments and suggestions. The manuscript in its current version can be endorsed for publication

7. PLOS authors have the option to publish the peer review history of their article (what does this mean?). If published, this will include your full peer review and any attached files.

Reviewer #1: No

Reviewer #2: **Yes**

---

## [Editor Report · Acceptance letter]

30 Jan 2024

PONE-D-23-31183R1 

PLOS ONE

Dear Dr. Mohebbi, 

I'm pleased to inform you that your manuscript has been deemed suitable for publication in PLOS ONE. Congratulations! Your manuscript is now being handed over to our production team.

Kind regards, 

on behalf of

Dr. Ahmed A. Al-Karmalawy 

Academic Editor

PLOS ONE